# A computational study of the Warburg effect identifies metabolic targets inhibiting cancer migration

Keren Yizhak[1,*,†], Sylvia E Le Dévédec[2,†], Vasiliki Maria Rogkoti[2], Franziska Baenke[3], Vincent C de Boer[4], Christian Frezza[5], Almut Schulze[3], Bob van de Water[2,‡] & Eytan Ruppin[1,6,‡,**]

## Abstract

Over the last decade, the field of cancer metabolism has mainly focused on studying the role of tumorigenic metabolic rewiring in supporting cancer proliferation. Here, we perform the first genome-scale computational study of the metabolic underpinnings of cancer migration. We build genome-scale metabolic models of the NCI-60 cell lines that capture the Warburg effect (aerobic glycolysis) typically occurring in cancer cells. The extent of the Warburg effect in each of these cell line models is quantified by the ratio of glycolytic to oxidative ATP flux (AFR), which is found to be highly positively associated with cancer cell migration. We hence predicted that targeting genes that mitigate the Warburg effect by reducing the AFR may specifically inhibit cancer migration. By testing the anti-migratory effects of silencing such 17 top predicted genes in four breast and lung cancer cell lines, we find that up to 13 of these novel predictions significantly attenuate cell migration either in all or one cell line only, while having almost no effect on cell proliferation. Furthermore, in accordance with the predictions, a significant reduction is observed in the ratio between experimentally measured ECAR and OCR levels following these perturbations. Inhibiting anti-migratory targets is a promising future avenue in treating cancer since it may decrease cytotoxic-related side effects that plague current anti-proliferative treatments. Furthermore, it may reduce cytotoxic-related clonal selection of more aggressive cancer cells and the likelihood of emerging resistance.

**Keywords** cancer cell migration; cellular metabolism; genome-scale metabolic modeling

**Subject Categories** Genome-Scale & Integrative Biology; Metabolism; Computational Biology
**Mol Syst Biol.** (2014) 10: 744

## Introduction

Altered tumor metabolism has become a generally regarded hallmark of cancer (Hanahan & Weinberg, 2011). The initial recognition that metabolism is altered in cancer can be traced back to Otto Warburg's early studies, showing that transformed cells consume glucose at an abnormally high rate and largely reduce it to lactate, even in the presence of oxygen (Warburg, 1956). Over the last decade, much of the field of cancer metabolism has focused on the role of the Warburg effect in supporting cancer proliferation (Vander Heiden *et al*, 2009). However, the role of this process in supporting other fundamental cancer phenotypes such as cellular migration has received far less attention.

Contemporary cytotoxic cancer treatment has been mainly based on drugs that kill proliferating cells generally unselectively and are therefore accompanied by many undesirable side effects. Drug targets that can inhibit migration but leave cellular proliferation relatively spared may be able to avoid such side effects. Such targets may have the additional benefit of reducing the selection for more resistant clones that occurs due to the elimination of treatment-sensitive cells. The growing availability of high-throughput measurements for a range of cancer cells presents an opportunity to study a wider scope of dysregulated metabolism across many different cancers. Here, we aim to

1   The Blavatnik School of Computer Science, Tel-Aviv University, Tel-Aviv, Israel
2   Division of Toxicology, Leiden Academic Centre for Drug Research, Leiden University, Leiden, The Netherlands
3   Gene Expression Analysis Laboratory, Cancer Research UK, London Research Institute, London, UK
4   Laboratory Genetic Metabolic Diseases, Academic Medical Center, Amsterdam, The Netherlands
5   MRC Cancer Unit, Hutchison/MRC Research Centre, University of Cambridge, Cambridge, UK
6   The Sackler School of Medicine, Tel-Aviv University, Tel-Aviv, Israel
    *Corresponding author. Tel: +972 3 6405378; E-mail: kerenyiz@post.tau.ac.il
    **Corresponding author. Tel: +972 3 6406528; E-mail: ruppin@post.tau.ac.il
    †These authors contributed equally to this study
    ‡These authors contributed equally to this study

integrate pertaining data with a genome-scale mechanistic model of human metabolism to study the role of the Warburg effect in tumor progression and its potential association with cellular migration.

Genome-scale metabolic modeling is an increasingly widely used computational framework for studying metabolism. Given the genome-scale metabolic model (GSMM) of a species alongside contextual information such as growth media and 'omics' data, one can obtain a fairly accurate prediction of numerous metabolic phenotypes, including growth rates, nutrient uptake rates, gene essentiality, and more (Covert *et al*, 2004). GSMMs have been used for various applications (Oberhardt *et al*, 2009; Chandrasekaran & Price, 2010; Jensen & Papin, 2010; Szappanos *et al*, 2011; Wessely *et al*, 2011; Lerman *et al*, 2012; Nogales *et al*, 2012; Schuetz *et al*, 2012) including drug discovery (Trawick & Schilling, 2006; Oberhardt *et al*, 2013; Yizhak *et al*, 2013) and metabolic engineering (Burgard *et al*, 2003; Pharkya *et al*, 2004). Over the last few years, GSMMs have been successfully used for modeling human metabolism as well (Duarte *et al*, 2007; Ma *et al*, 2007; Shlomi *et al*, 2008; Gille *et al*, 2010; Lewis *et al*, 2010; Mardinoglu *et al*, 2013). Specifically, GSMM models of cancer cells have been reconstructed and applied for predicting selective drug targets, as well as for studying the role of tumor suppressors and oxidative stress (Folger *et al*, 2011; Frezza *et al*, 2011; Agren *et al*, 2012, 2014; Jerby *et al*, 2012; Goldstein *et al*, 2013; Gatto *et al*, 2014). In the context of studying the Warburg effect, the original human metabolic model does not predict forced lactate secretion under maximal biomass production rate, even when oxygen consumption rate equals zero. This renders it unsuitable for studying the Warburg effect as is, as already noted by (Shlomi *et al*, 2011). While the addition of solvent capacity constraints has been shown to overcome this hurdle in principle (Shlomi *et al*, 2011), this addition requires enzymatic kinetic data which are still largely absent on a genome-scale.

In this study, we utilize individual genome-scale metabolic models tailored separately to each of the NCI-60 cancer cell lines to study the role of the Warburg effect in supporting cancer cellular migratory capacity. We first test and validate the individual models against both existing and novel bioenergetic experimental data. Then, we examine the extent of the Warburg effect occurring in a given cancer cell line, by quantifying the glycolytic to oxidative ATP flux ratio (AFR). We find that the AFR is highly positively correlated with cancer cell migration, emphasizing the role of glycolytic flux in supporting the more aggressive metastatic stages of tumor development. To determine whether a causal relation exists between AFR levels and cell migration, we predict gene silencing that reduce this ratio. These potential targets are then filtered further to exclude those predicted to result in cell lethality. Reassuringly, the predicted targets are found to be significantly more highly expressed in metastatic and high-grade breast cancer tumors. Experimental investigation of the top predicted targets via siRNA-mediated knockdown shows that a significant portion of them truly attenuate cancer cell migration without inducing a lethal effect. Furthermore, in accordance with the predictions, a significant reduction is observed in the ratio between ECAR and OCR levels following these genes silencing perturbations.

# Results

## Stoichiometric and flux capacity constraints successfully capture the coupling of high cell proliferation rate to lactate secretion across individual NCI-60 cancer models

As a starting point for this study, we developed a set of metabolic models specific for each of the NCI-60 cell lines. We built these models using a new algorithm we have recently developed termed PRIME, for building individual models of cells from pertaining omics data (Yizhak *et al*, submitted, Supplementary Information and Supplementary Fig S1). PRIME uses the generic human model as a scaffold and sets maximal flux capacity constraints over a subset of its growth-associated reactions according to the expression levels of their corresponding catalyzing enzymes in each of the target cell lines.

An important hallmark of cancerous cells is the production of lactate through the Warburg effect (Warburg, 1956). As a first step in validating the basic function of our NCI-60 models, we assessed whether maximizing biomass forces production of lactate, which would signify proper coupling of biomass production with lactate output as seen in cancer cells. We found that the models indeed must secrete lactate under biomass maximization (Supplementary Information and Supplementary Fig S2). Hence, in contrast to the original generic model of human metabolism, they enable us to systematically assess the extent of lactate secretion and study the Warburg effect across a wide range of cancer cell lines without needing to add (mostly unknown) solvent capacity constraints, thus identifying its functional correlates on a genome scale.

## Comparing predicted versus experimentally measured bioenergetics capacity

We compared the predicted lactate secretion rates across all cell lines to those measured experimentally by Jain *et al* (Jain *et al*, 2012), obtaining a moderate but significant correlation (Spearman correlation $R = 0.36$, $P$-value $= 5.7e-3$, Fig 1A, Materials and Methods). To further test the models' performance under different environmental conditions, we measured lactate secretion rates in four breast cancer cell lines, T47D, MCF7, BT549, and Hs578T (Supplementary Dataset S1), under both normoxic and hypoxic conditions (see Materials and Methods). Utilizing the corresponding cell line models from the NCI-60 set, we found a high correlation between measured and predicted lactate secretion levels across both conditions (Spearman correlation $R = 0.95$, $P$-value $= 1.1e-3$, Fig 1B).

The ratio of glycolytic versus oxidative capacity in a cell can be quantified using its extracellular acidification rate (ECAR, a proxy of lactate secretion) and its oxygen consumption rate (OCR). To further examine how well our cell line models capture measured Warburg-related activity in response to genetic perturbations, we utilized measured ECAR and OCR levels in response to perturbations in two NCI-60 lung cancer cell lines (A549 and H460), and compared the results to predictions from our models (Materials and Methods) (Wu *et al*, 2007). Qualitatively similar ECAR and OCR changes are found in response to various enzymatic perturbations along the glycolytic pathway. Specifically, increased glycolytic inhibition resulted in reduced ECAR and elevated OCR levels in both cells, while the maximum cellular respiration increase in H460 cells

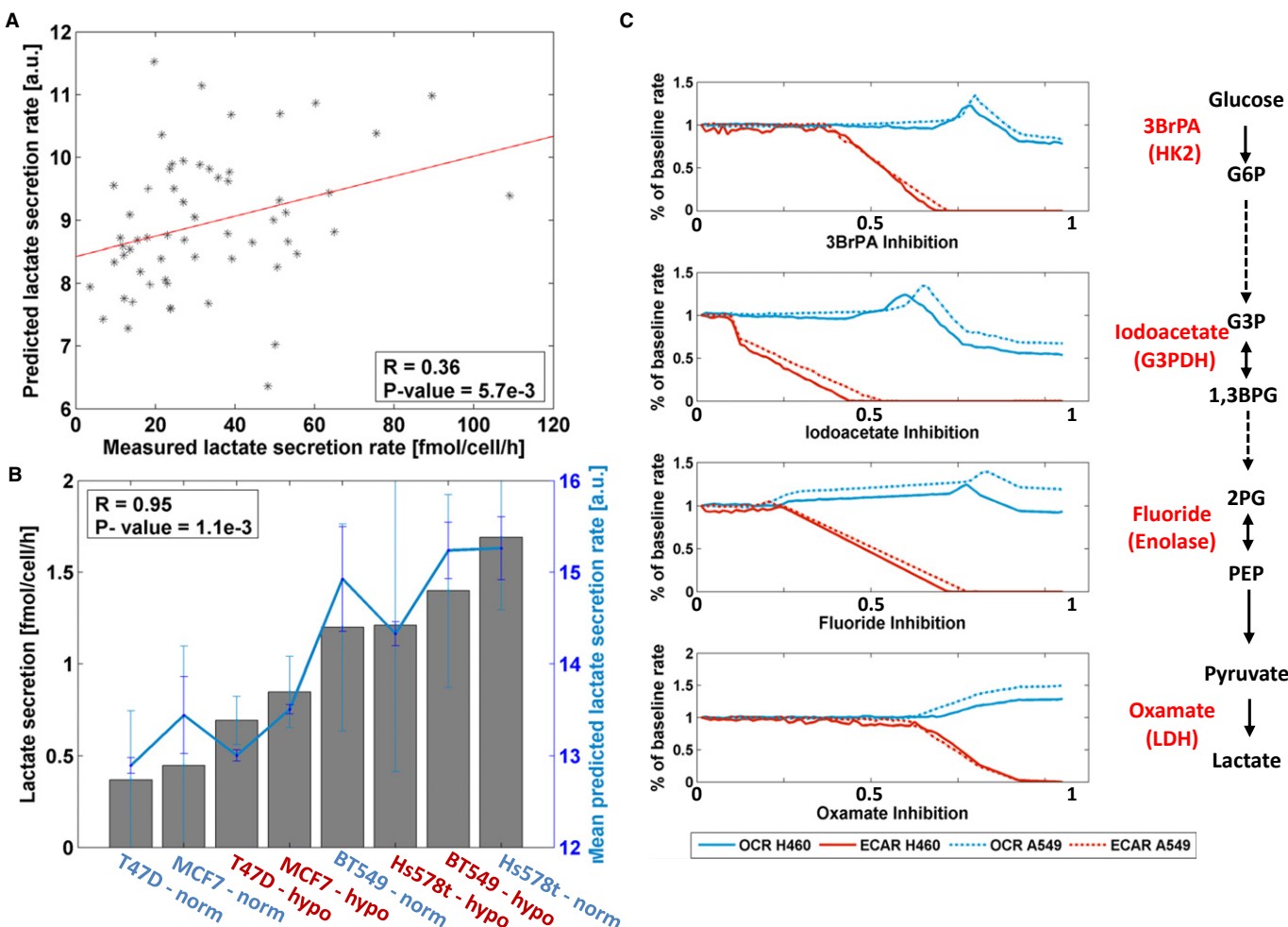

**Figure 1.  A comparison between experimental and predicted *in silico* measurements of lactate secretion (or ECAR) and OCR across different cancer cell lines.**

A   Measured versus predicted lactate secretion rates across the 59 cell lines available at Jain *et al* (2012).

B   Measured versus predicted lactate secretion rates in hypoxic (red) and normoxic (blue) conditions for four breast cancer cell lines: T47D, MCF7, BT549, and Hs578T. Bars represent the measured lactate secretion rates and the line represents the corresponding predicted rates. Error bars represent SD; number of samples for experimental data (bars) is *n* = 7; number of samples for predicted data (line) is *n* = 1000.

C   Predicted ECAR and OCR by the A549 and H460 cell line models following inhibitory perturbations in the glycolytic pathway. The models predictions show a decrease in ECAR (red line) and an increase in OCR (blue line). As found experimentally, the predicted OCR increase in H460 cells is lower than that found for A549 cells. The *x*-axes represent the level of inhibition imposed, starting from a zero to a maximal inhibition (Materials and Methods). The specific perturbations include 3BpRA that inhibits the enzyme hexokinase 2; Iodoacetate that inhibits the enzyme glycerol-3-phosphate dehydrogenase; Fluoride that inhibits the enzyme enolase; and Oxamate that inhibits the enzyme lactate dehydrogenase.

observed after all glycolysis inhibitors was lower than the corresponding increase in A549 cells (Fig 1C).

**Quantifying the Warburg effect and its relation to proliferation and migration across the NCI-60 cell lines**

While ECAR and OCR are the commonly used measures for experimentally quantifying the bioenergetic capacity of the cell and thus the Warburg effect, the genome-wide scope of GSMMs enables us to examine other putative measures as well. One promising such measure we examined is the ratio between the ATP flux rate in the glycolysis versus its flux rate in OXPHOS (AFR). Clearly, higher AFR values denote more 'Warburgian' cell lines and vice versa. A comparison of our new AFR metric versus the aforementioned

state-of-the-art ECAR/OCR ratio (EOR) (Materials and Methods and Supplementary Dataset S2) showed a significant correlation across the NCI-60 models (Spearman correlation *R* = 0.66, *P*-value = 2e−8). Testing both measures using a genome-wide NCI-60 drug response dataset (Scherf *et al*, 2000), we find that the model-predicted wild-type AFR levels across all cell line models are significantly correlated (Spearman *P*-value < 0.05; FDR corrected with α = 0.05) with Gi50 values of 30% of the compounds across these cell lines (empiric *P*-value < 9.9e−4), whereas the model-predicted EOR measure accomplish this task for only 19% of the compounds (Materials and Methods). Interestingly, we find that out of the 30% AFR-Gi50-correlated compounds, 97% are positively correlated, suggesting that the more 'Warburgian' cell lines are less responsive and therefore require higher dosage of compound to suppress their

growth. The effect of most of these compounds is also negatively correlated with the cells' growth rates, suggesting that slowly proliferating cells are more resistant to treatment (similar results were previously shown for compounds targeting cell growth (Penault-Llorca *et al*, 2009; Vincent-Salomon *et al*, 2004)). Interestingly, the response to many compounds in this dataset shows a significant association with the AFR measure while having no association with the cells' growth rate. 133 such compounds were identified (Supplementary Dataset S3), possibly suggesting that their mechanism might be related to the Warburg level of the cells rather than to their proliferation. Finally, predicted AFR values correctly separate between epithelial and mesenchymal breast cancer cell lines (with

the more aggressive mesenchymal cell lines exhibiting larger Warburg effect (Sarrio *et al*, 2008), Fig 2A). Once again, the AFR was more predictive of this experimental observation than the EOR (Supplementary Dataset S2).

We next turned to our primary objective of examining the relation between the Warburg effect and tumor proliferation and migration. To this end, we experimentally measured the migration speed of six NCI-60 breast cancer cell lines (Fig 2B and C, Materials and Methods, Supplementary Fig S3, and Supplementary Dataset S2) and utilized publically available measured growth rates for these cell lines. While the AFR correlates markedly negatively with cell growth rate (Spearman correlation of $R = -0.55$, *P*-value = 4.53e−6,

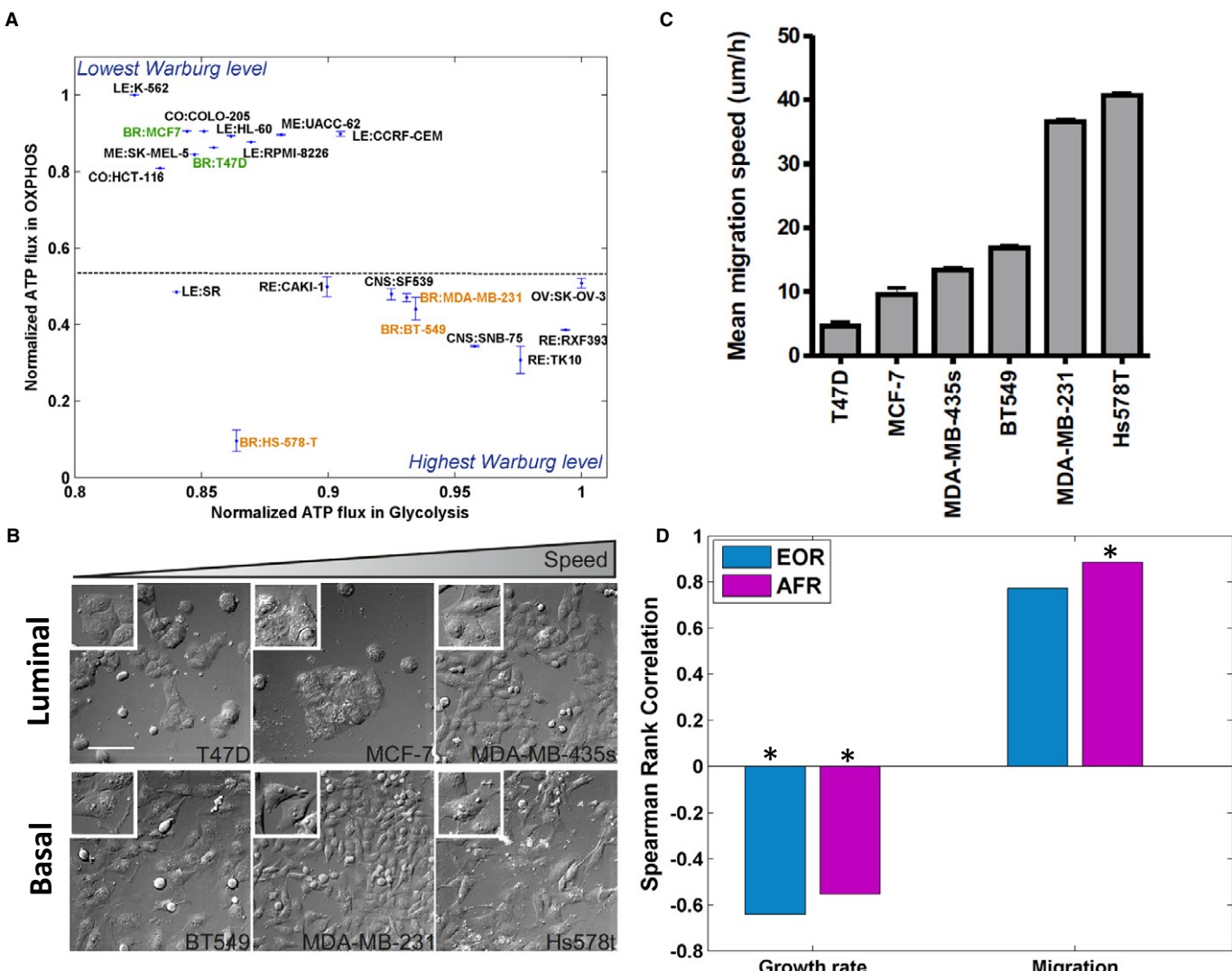

**Figure 2. Association between AFR levels and cell proliferation and migration.**

A   The 20 cell lines that are predicted to exhibit the Warburg effect to the greatest/least extent according to the AFR measure. The *x*-axis and *y*-axis represent the mean and SD of the normalized ATP flux rate in glycolysis and OXPHOS, respectively (Materials and Methods). The AFR measure correctly separates between mesenchymal (orange) and epithelial cell lines (green), showing that the former (which are known to be more aggressive) have higher AFR levels.

B   We analyzed a panel of six breast cancer cell lines for their migration capacity using live cell imaging. Differential Interference Contrast (DIC) images of the six cell lines in the order of their respective migration speed (from low to high), scale bar is 100 μm (Materials and Methods).

C   The average migration speed of cells followed for 12 h in complete medium. Error bars represent SEM; the number of samples is between *n* = 100 and *n* = 200.

D   The correlation of predicted model-based EOR and AFR measures to growth and migration rates measured experimentally. Both measures represent a negative correlation with growth and a positive correlation with migration rates. Significant results (*P*-value < 0.05) are marked with an asterisk.

                    

Fig 2D and Supplementary Table S1), it correlates even more strongly in the positive direction with cancer cell migration (Spearman correlation of $R = 0.88$, $P$-value $= 0.03$, Fig 2D and Supplementary Table S1). Controlling for the cell lines' measured growth rates, this correlation becomes even more significant (partial Spearman correlation of $R = 0.96$, $P$-value $= 7e-3$, Supplementary Table S1). Overall, this finding suggests that glycolytic flux correlates with migration rather than with growth, while OXPHOS flux exhibits the opposite behavior. A similar association between lactate secretion and growth rate has been recently found in an experimental study by Jain *et al* (Jain *et al*, 2012) across the entire NCI-60 collection (Spearman correlation of $R = -0.22$, $P = 0.09$). Furthermore, previous studies have shown that high concentrations of lactate correlate with a high incidence of distant metastasis (Hirschhaeuser *et al*, 2011). The overall picture portrayed by these correlations is that while glycolytic carbon diverted to biosynthetic pathways may support cell proliferation, non-diverted glycolytic carbon supports cell migration and metastasis (Supplementary Fig S4).

## Predicting drug targets that revert the AFR and hence may inhibit cancer migration

The congruence between AFR levels and disease severity led us to ask if we could build upon this association to identify potential new drug targets. We searched for drug targets predicted to reduce the AFR ratio by simulating the knockout of each metabolic reaction across the NCI-60 models, and examining the effects of the knockouts on biomass production, lactate secretion, and the AFR. As lactate secretion is a basic indicator of the Warburg effect, we first identified a set of 113 reactions whose knockout is predicted to abolish lactate secretion rate in all cancer cell lines under biomass maximization. Interestingly, the set of enzymes catalyzing these reactions is significantly more highly expressed in the NCI-60 cell lines than the background metabolic genes (one-sided Wilcoxon $P$-value $< 1.6e-8$), indicating the potential oncogenic nature of these genes.

   To avoid selecting for drug-resistant clones it would be advantageous to develop drugs that reduce the virulence of cancer cells but avoid killing them. The knockout of 12 of 113 lactate-reducing reactions reduces the AFR but relatively spares biomass production (Materials and Methods and Supplementary Table S2). Importantly, the knockout of these 12 reactions according to models of healthy lymphoblast cells built by PRIME (Choy *et al*, 2008) also spares their biomass production (Materials and Methods). Moreover, we found that none of the lymphoblast cell lines show the forced lactate secretion that is observed in cancer cells. While the Warburg effect is sometimes referred in the literature as occurring in highly proliferating cells in general, our analysis finds that this phenomenon is apparently more prominent in cancer cells, at least with regard to the lymphoblastoid cell population studied here.

   The final list of predicted gene targets includes 17 metabolic enzymes that are associated with the final 12 reactions, spanning glycolysis, serine, and methionine metabolism (Fig 3A). 10 of the predicted targets have significantly higher expression levels in metastatic versus non-metastatic breast cancer patients (Chang *et al*, 2005) (one-sided Wilcoxon $P$-value $< 0.05$, Fig 3B). Moreover, 9 of the predicted targets exhibit higher expression levels in grade 3 tumors than in grade 1 tumors (Miller *et al*, 2005) (one-sided

Wilcoxon $P$-value $< 0.05$, Fig 3C). Finally, lower expression of nine of the predicted targets is significantly associated with improved long-term survival (Curtis *et al*, 2012) (log-rank $P$-value $< 0.05$, Fig 3D), testifying for their potential role as therapeutic targets. All $P$-values are corrected for multiple hypothesis using FDR with $\alpha = 0.05$.

## siRNA-mediated gene knockdown experiments testing the predicted targets

To experimentally test our predictions we silenced the 17 predicted AFR-reducing genes and examined their phenotypic effects in the MDA-MB-231, MDA-MB-435, BT549, and A549 cell lines. Knockdown experiments were performed with SmartPools from Dharmacon using a live cell migration and fixed proliferation assays (Materials and Methods). 8–13 out of the 17 enzymes (8–10 out of 12 metabolic reactions) were found to significantly attenuate migration speed in each cell line (two-sided $t$-test $P$-value $< 0.05$, FDR corrected with $\alpha = 0.05$, Fig 4, Materials and Methods and Supplementary Dataset S4). This result is highly significant as only 17% of the metabolic genes were found to impair cell migration in a siRNA screen of 190 metabolic genes (Fokkelman M, Rogkoti VM *et al*, unpublished data, Bernoulli $P$-value in the range of $3.9e-3$ and $1.18e-7$). Of note, the association between the gene expression of the predicted targets and the measured migration speed is insignificant for all targets but one, testifying for the inherent value of our model-based prediction analysis (Supplementary Table S3). It should also be noted that the knockdown of the three splices of the enolase gene have almost no significant effect on these cells' migration speed, possibly because of isoenzymes backup mechanisms. Importantly, most of the gene knockdown experiments do not manifest any significant effects on cell proliferation (Fig 4). In accordance with the findings of Simpson *et al* (Simpson *et al*, 2008), we found that the correlation between the reduction in migration speed and reduction in proliferation rate is mostly insignificant (Supplementary Dataset S4), suggesting that the reduced migration observed is not simply a consequence of common mechanisms hindering proliferation, but rather that it occurs due to the disruption of distinct migratory-associated metabolic pathways.

## ECAR and OCR levels following selected gene silencing

To further study the association between reduced AFR levels and impaired cell migration we used the Seahorse XF96 extracellular flux analyzer to measure both ECAR and OCR fluxes in the MDA-MB-231 cell line, following knockdown of a selected group of targets (Materials and Methods and Supplementary Fig S6). As the AFR measure is very difficult to measure experimentally, we tested the conventionally measured EOR (ECAR/OCR) as its proxy. We focused on a subset of seven genes (Fig 5) whose knockdown is predicted to have the highest effect on cell migration and span all three predicted metabolic pathways. As shown in Fig 5, a significant EOR reduction versus the control is found for all seven examined genes (two-sided $t$-test $P$-value $< 0.05$, FDR corrected with $\alpha = 0.05$, Materials and Methods and Supplementary Table S4). The silencing of the four glycolytic genes (*HK2*, *PGAM1*, *PGK2*, and *GAPDH*) results in both decreased ECAR and increased OCR levels, while the silencing of the serine- and methionine-associated genes

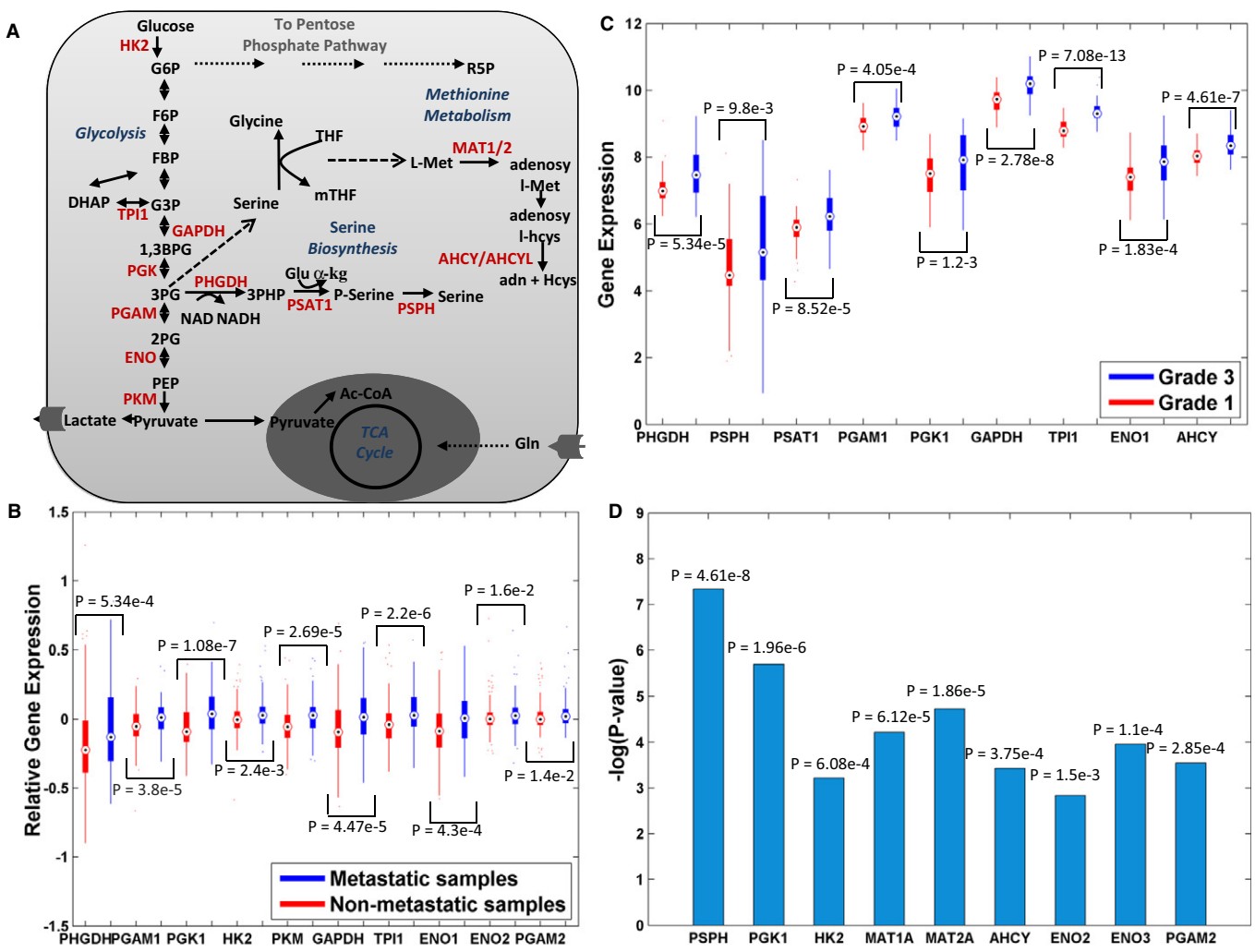

**Figure 3.  Gene targets that are predicted to reduce the AFR and their association with prognostic markers of breast cancer patients.**

A    A schematic representation of the 12 predicted gene targets, marked in red.

B    Ten predicted targets that show a significantly higher expression in metastatic versus non-metastatic tumor samples ($n$ = 295).

C    Nine predicted targets that show a significantly higher expression in grade 3 versus grade 1 tumor samples ($n$ = 236).

D    Nine predicted targets whose lower expression is significantly associated with improved long-term survival ($n$ = 1568).

(*PSPH*, *AHCY*, and *PHGDH*) results with decreased ECAR solely (Fig 5A). Furthermore, a matching significant difference in experimentally measured EOR levels is found between the lowest and highest AFR-reducing genes (one-sided Wilcoxon *P*-value = 0.05). Overall, taken together our results testify that, as predicted, the knockdown of the top-ranked genes results in attenuated cell migration that is accompanied by reduced EOR and AFR levels.

## Discussion

In this study we explored the role of the Warburg effect in supporting tumor migration, going beyond recent investigations focusing on its role in assisting cancer proliferation. A model-based investigation across cancer cell lines shows that the ratio between glycolytic and oxidative ATP flux rate is significantly associated with cancer migratory behavior. Gene silencing perturbations predicted to reduce this

ratio were indeed found to attenuate cell migration, and result with a significant reduction in ECAR to OCR levels. Of note, our modeling approach relies on gene expression differences between the cells and does not take into account specific uptake rates. It is therefore more suited for capturing qualitative rather than exact quantitative differences between the cells, as demonstrated throughout the paper. Moreover, the lion share of our analysis is focused on the simulations of perturbations where specific uptake rates are not available. Nonetheless, utilizing such uptake measurements can significantly increase the correlation to the measured lactate rates (Spearman correlation $R$ = 0.67, *P*-value = 1.5e−8), suggesting that uptake rates measurements under perturbation states can significantly increase the models' prediction power.

Our AFR measure is conceptually analogous to a bioenergetic (BEC) index previously introduced by Cuezva *et al* (Cuezva *et al*, 2002). In that study, the ratio between the expression of the glycolytic enzyme glyceraldehyde-3-phosphate dehydrogenase (GAPDH)

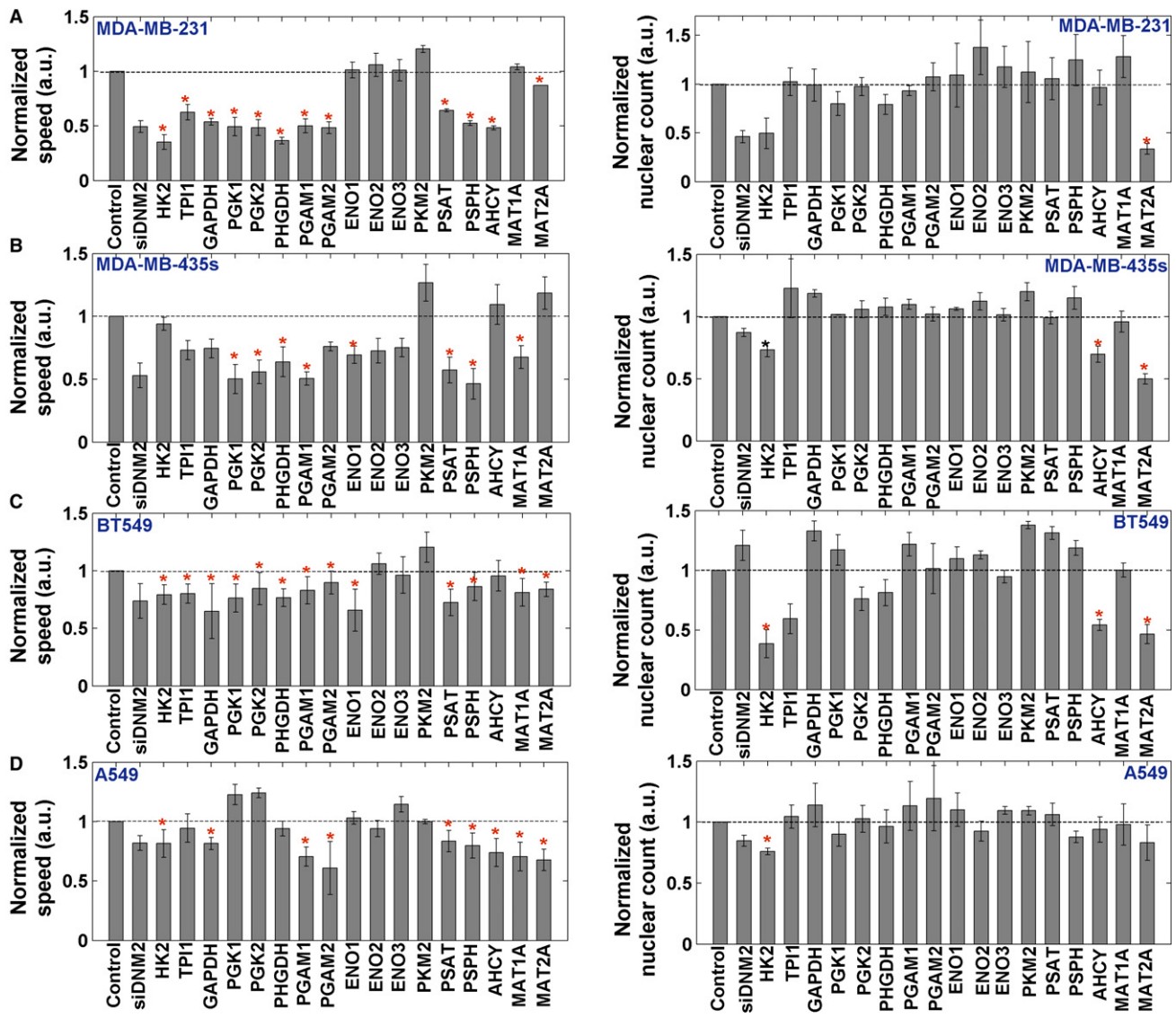

**Figure 4.  Normalized to control mean speed per SmartPool gene silencing of the predicted targets.**

A–D   The four different cell lines that were analyzed: MDA-MB-231, MDA-MB-435s, BT549, and A549. Significant results (two-sided *t*-test, *P*-value < 0.05 after correcting for multiple hypothesis using FDR with α = 0.05) are marked with an asterisk. Two different controls are used: (1) non-targeting siRNA (= negative control); and (2) a positive control DNM2 which is known to block both migration and proliferation (Ezratty *et al*, 2005). Left panel shows migration speed and right panel shows nuclear count. Error bars represent SD; the number of samples is *n* = 3.

and the β-catalytic subunit of ATP synthase forming the BEC index was found to have a prognostic value in assessing the clinical outcome of patients with early-stage colorectal carcinomas. The AFR measure and the BEC index (as computed by its corresponding RNA levels) are significantly correlated (Spearman *R* = 0.58, *P*-value = 1.6e−6) across the NCI-60 cell lines, and the BEC index is perfectly correlated with migration speed across the six breast cancer cell lines (Spearman *R* = 1, *P*-value = 2.8e−3). However, the BEC index has inferior performance in predicting drug response (Supplementary Table S1).

The finding that enhanced glycolytic activity plays a key role in cancer cell migration is also in line with a very recent study by De Bock *et al*, showing that glycolysis is the major source of ATP

production in endothelial cells and that the silencing of the glycolytic regulator PFKFB3 impairs the cell migration capacity and interferes with vessel sprouting (De Bock *et al*, 2013). In addition, silencing of PFKFB3 was shown to suppress cell proliferation in about 50% (De Bock *et al*, 2013). Overall, the results presented in this study, as well as findings reported by others (Simpson *et al*, 2008), suggest that proliferation and migration are not mutually exclusive, and the effect of potential targets on both processes should be carefully examined.

Some of our predicted targets have been previously studied in the context of cell proliferation as well (Cheong *et al*, 2012). Possemato *et al* (Possemato *et al*, 2011) have showed that suppression of *PHGDH* in cell lines with elevated PHGDH expression, but not

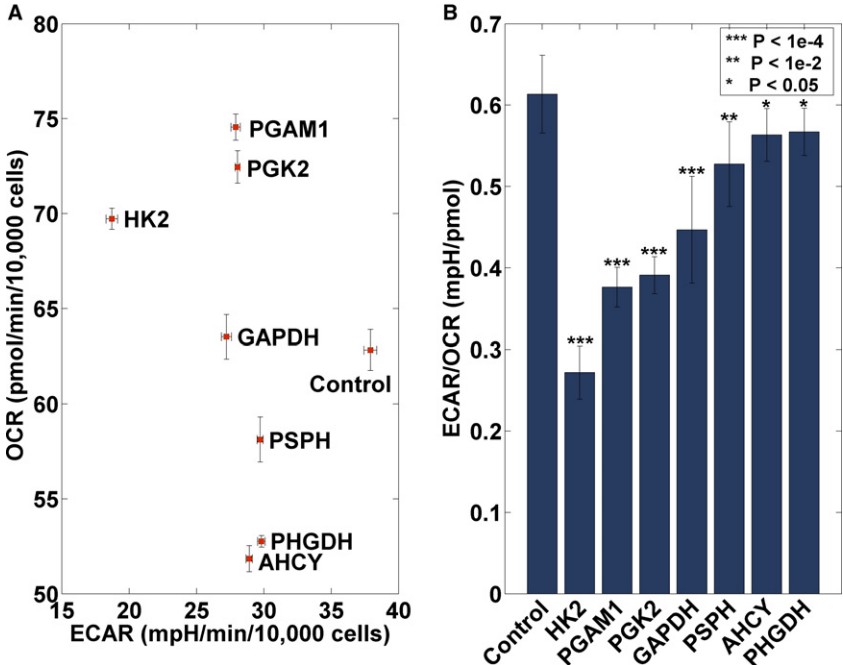

**Figure 5.  ECAR and OCR levels of top predicted gene targets.**

A  Mean and SEM (normalized to nuclear count) ECAR and OCR levels after silencing of seven different genes (*HK2*, *PGAM1*, *PGK2*, *GAPDH*, *PSPH*, *AHCY*, and *PHGDH*) compared to the control. Silencing of the four glycolytic genes results in both a decrease in ECAR levels (*x*-axis) and an increase in OCR levels (*y*-axis), while the serine- and methionine-associated genes show only a decrease in ECAR levels. Error bars represent SEM. The number of samples is *n* = 18.

B  Mean and SD of computed ECAR/OCR (EOR) levels for control and selected gene silencing (Materials and Methods). For all genes a significant reduction in EOR levels is observed. Error bars represent SD. The number of samples is *n* = 18.

in those without, inhibits cell proliferation. Accordingly, as *PHGDH* is not amplified in the cell line MDA-MB-231 which was examined in both studies, its suppression is indeed non-lethal. However, we show that its suppression significantly attenuates cell migration, suggesting that metabolic enzymes can promote different cancerous phenotypes in different cancer cells.

Remarkably, analyzing the model-predicted flux rates has successfully uncovered a fundamental association between the AFR and cancer migration, even given the relatively small set of cell lines for which migration was measured. Our analysis has also revealed other potential associations between individual fluxes and cell migration (Supplementary Fig S4). However, future studies measuring cellular migration data across a much wider array of cell lines (of the order for which we already have proliferation data) are needed to determine the actual significance of these potential leads. As this study has shown, cellular proliferation and migration have distinct underlying metabolite correlates; understanding the metabolic correlates that are strongly associated with cell migration may lead to new anti-metastatic treatment opportunities. It is important to note, however, that while the inhibition of migration alone might be a good strategy for avoiding the adverse side effects of cytotoxic treatment, cell migration is a crucial process also in normal physiology, for instance, in immune response and tissue repair (Förster *et al*, 1999; Ridley *et al*, 2003). Therefore, future anti-migratory drugs may pose different drug selectivity challenges that should be carefully addressed in the future studies. Irrespectively, they may result in lesser clonal selection, and as a result, their usage may be accompanied with lesser rate of emergence of drug-resistant clones.

# Materials and Methods

## Computational methods

### Genome-scale metabolic modeling (GSSM)

A metabolic network consisting of *m* metabolites and *n* reactions can be represented by a *stoichiometric matrix S*, where the entry $S_{ij}$ represents the stoichiometric coefficient of metabolite *i* in reaction *j* (Price *et al*, 2004). A CBM model imposes mass balance, directionality, and flux capacity constraints on the space of possible fluxes in the metabolic network's reactions through a set of linear equations:

$$Sv = 0 \qquad (1)$$

$$v_{min} \leq v \leq v_{max} \qquad (2)$$

where *v* stands for the flux vector for all of the reactions in the model (i.e. the *flux distribution*). The exchange of metabolites with the environment is represented as a set of *exchange (transport) reactions*, enabling a pre-defined set of metabolites to be either taken up or secreted from the growth media. The steady-state assumption represented in equation (1) constrains the production rate of each metabolite to be equal to its consumption rate. Enzymatic directionality and flux capacity constraints define lower and upper bounds on the fluxes and are embedded in equation (2). In the following, flux vectors satisfying these conditions will be referred to as feasible steady-state flux distributions. Gene knockouts are simulated by constraining the flux through the

corresponding metabolic reaction to zero. The biomass function utilized here is taken from (Folger *et al*, 2011). The media simulated in all the analyses throughout the paper is the RPMI-1640 media that was used to grow the cell lines experimentally (Lee *et al*, 2007; Choy *et al*, 2008).

*Building cell-specific metabolic models and computing lactate secretion*
Our method to reconstruct the NCI-60 cancer cell lines (see Supplementary Material, based on the yet unpublished methods in Yizhak *et al*, submitted) required several key inputs: (a) the generic human model (Duarte *et al*, 2007), (b) gene expression data for each cancer cell line from (Lee *et al*, 2007), and (c) growth rate measurements. The algorithm then reconstructs a specific metabolic model for each sample by modifying the upper bounds of growth-associated reactions in accordance with their gene expression (Note: the growth rates were used only to determine which reactions should be used in constraining the models, in order to obtain models that were as physiologically relevant as possible; they were not used to determine reaction bounds). A similar procedure was used to reconstruct the lymphoblast metabolic models (Choy *et al*, 2008) for comparison against normal proliferating cells. A more detailed description is found in the Supplementary Material.

Simulations of the Warburg effect include the examination of minimal lactate production rate under different demands for biomass production, glucose, glutamine, and oxygen uptake rates (Supplementary Material). We examined the minimal value of lactate secretion as it testifies whether or not the cell is enforced to secrete lactate under a given condition (Supplementary Fig S1). All the correlations reported in the paper are Spearman rank correlations and their associated *P*-values are computed using the exact permutation distribution.

*Calculating wild-type and perturbed lactate secretion rates and OCR levels*
For simulating lactate secretion under normoxic conditions (when comparing to Jain *et al* (Jain *et al*, 2012), Wu *et al* (Wu *et al*, 2007) and the breast cancer data collected in this paper), oxygen maximal uptake rate was set to the highest value under which minimal lactate secretion is positive. Since metabolic models are designed to maximize growth yield rather than growth rate, using an unlimited amount of oxygen in GSMM simulations will result in a state where the minimal lactate secretion rate equals zero. However, it's important to note that even under the limited oxygen levels simulated here, the generic human model doesn't show lactate secretion (as opposed to the NCI-60 cancer cell line models described above). For simulating the hypoxic conditions measured here for the breast cancer cell lines, we lowered the oxygen maximal uptake rate by 50% of its normoxic state as described above. Under each of these conditions, we sampled the solution space under maximal biomass yield and obtained 1,000 feasible flux distributions (Bordel *et al*, 2010). The predicted lactate secretion rate is the average lactate secretion flux over these samples. For emulating the perturbation experiments in Wu *et al* we gradually lowered the bound of the corresponding compound target (from the maximal bound to 0) and repeated the procedure described above for computing the ECAR (lactate secretion) and the OCR, which in a similar manner is defined as the average oxygen consumption flux across all samples.

*Calculating the EOR and AFR measures for assessing the Warburg level of the cell lines and using them to predict drug response*
The EOR and AFR measures were calculated in a similar manner to that described above. Specifically, the EOR is calculated as the mean over lactate secretion across all samples divided by the mean over oxygen consumption across all samples. Similarly, the AFR is calculated as the mean flux carried by the reactions producing ATP in glycolysis versus the mean flux carried by the reaction producing ATP in OXPHOS. To determine an empiric *P*-value in the drug response analysis we randomly shuffled the drug response data 1,000 times, each time examining the resulting Wilcoxon *P*-value over the original set of cell lines.

*Predicting the effect of reaction knockouts*
Each metabolic reaction in each cell line model is perturbed by constraining its flux to zero. Under each perturbation the minimal lactate secretion (under maximal growth rate) and the maximal growth rate is calculated. The set of reactions that eliminate forced lactate secretion while maintaining a level of cell growth that is > 10% of the wild-type growth prediction is further tested for the AFR level. The mean AFR level for each cell line under each of these perturbations is calculated over 1,000 flux distribution samples as described above. The final set of predicted reactions includes those whose knockout reduces the AFR to below 60% of its wild-type level.

*Datasets*
Growth rate measurements and drug response data were downloaded from the NCI website.

Growth rate: http://dtp.nci.nih.gov/docs/misc/common_files/cell_list.html

Drug response: http://discover.nci.nih.gov/nature2000/natureintromain.jsp

## Experimentally measuring lactate secretion rates of breast cancer cell lines

*Cell Culture*
The MCF7, T47D, Hs578T and BT549 breast cancer cell lines were obtained from the American Type Culture Collection and London Research Institute Cell Services. Cells were cultured in DMEM/F12 (1:1), with 2 mM L-glutamine and penicillin/streptomycin. Medium was supplemented with 10% FCS (GIBCO) for the cancer cell lines and 5% horse serum, 20 ng/ml EGF, 5 μg/ml hydrocortisone, 10 μg/ml insulin, and 100 ng/ml cholera toxin for the non-malignant cell lines.

*Lactate secretion measurements*
Cells were cultured under normoxic (20% $O_2$) and hypoxic (0.5% $O_2$) conditions for 72 h. Cells were starved of glucose and glutamine for 1 h and full medium was added for 1 h. Lactate secretion was determined from normoxic and hypoxic cells and normalized to cell growth (increase in total protein during the 72 h incubation in normoxia). Lactate concentrations in media incubated with or without cells were determined using lactate assay kits (BioVision). Total protein content determined by Sulforhodamine B assay was used for normalization. Two experiments were performed with three or four biologically independent replicates (total of seven replicates).

## Cell culture for live cell imaging and cell migration assays

T47D, MCF-7, MDA-MB-435, BT549, MDA-MB-231 and Hs578t were cultured in RPMI (GIBCO, Life Technologies, Carlsbad, CA, USA) supplemented with 10% FBS (PAA, Pashing Austria) and 100 International Units/ml penicillin and 100 µg/ml streptomycin (Invitrogen, Carlsbad, CA, USA).

## Gene silencing

Human siRNA SmartPools (a combination of four individual singles) for the 17 predicted genes were purchased in siGENOME format from Dharmacon (Lafayette, CO, USA). Plates were diluted to 1 µM working concentration in complementary 1× siRNA buffer in a 96-well plate format. A non-targeting siRNA was used as negative control. A 50 nM reverse transfection was performed according to manufacturer's guidelines. Complex time was 20 min and 5,000 cells were added. The plate was placed in the incubator overnight and the medium was refreshed the following morning. After 48–72 h cells were used for various assays. Cell migration and metabolic flux assay experiments were performed in duplicate while the cell proliferation assay was performed in triplicate.

## Live cell imaging random cell migration assay

Glass bottom 96-well plates (Greiner Bio-one, Monroe, NC, USA) were coated with 20 µg/µl collagen type I (isolated from rat tails) for 1 h at 37°C. 48 h after silencing, the MDA-MB-231 cells were re-plated onto the collagen-coated glass bottom plate. 24 h after seeding, cells were pre-exposed for 45 min to 0.1 µg/µl Hoechst 33342 (Fisher Scientific, Hampton, NH, USA) to visualize nuclei. After refreshing the medium, cells were placed on a Nikon Eclipse TE2000-E microscope fitted with a 37°C incubation chamber, 20× objective (0.75 NA, 1.00 WD) automated stage and perfect focus system. Three positions per well were automatically defined, and the Differential Interference Contrast (DIC) and Hoechst signals were acquired with a CCD camera (Pixel size: 0.64 µm) every 20 min for a total imaging period of 12 h using NIS software (Nikon). All data were converted and analyzed using custom-made ImagePro Plus macros (Roosmalen *et al*, 2011). Cell migration was quantified by tracking nuclei in time. Changes in migration speed per knockdown were evaluated via a two-sided *t*-test comparing the speed for every individual cell followed overtime for 16 h and the corresponding control values. Data shown are normalized to control and represent only one replicate. Of note, for all four cell lines both replicates showed a $R^2$ of reproducibility above 0.75. Genes achieving *P*-value < 0.05 after correcting for multiple hypothesis using FDR with α = 0.05 are considered as hits.

## Proliferation assay

Cells were directly transfected and plated onto micro-clear 96-well plates (Greiner Bio-one). After 5 days of incubation, the cells were stained with Hoechst 33342 and fixed with TCA (Trichloroacetic acid) allowing both a nuclear counting and/or Sulforodamine B (SRB) readout. Whole wells were imaged using epi-fluorescence and the number of nuclei was determined using a custom-made ImagePro macro. Plates were further processed for SRB staining as described earlier (Zhang *et al*, 2011). SRB data showed a complete overlap with the nuclear count so this measure is used in all figures. Changes in proliferation rates upon knockdown when compared to control were evaluated in triplicate via a two-sided *t*-test. The mean proliferation rate after knockdown between all three replicates was calculated and normalized to the non-targeting siRNA (= control). Genes achieving *P*-value < 0.05 after correcting for multiple hypothesis using FDR with α = 0.05 are considered as hits.

## Metabolic flux assay

The bioenergetics flux of cells in response to gene silencing was assessed using the Seahorse XF96 extracellular flux analyzer (Seahorse Bioscience). About 8,000 MDA-MB-231 cells per well (Seahorse plate) were treated with siRNAs or control for 72 h. Each gene (in total 7) was knockdown in six different wells and the experiment was performed twice (so a total of six replicates per plate and two plates). Prior to measurement, the medium was replaced with unbuffered DMEM XF assay medium. The basal oxygen consumption rate (OCR) and extracellular acidification rate (ECAR) were then determined using the XP96 plate reader with the standard program as recommended by the manufacturer: three measurements per well were done (so for each gene 18 measurements were obtained for both OCR and ECAR). After the measurements were completed, the plates were live stained with Hoechst 33342 for 1 h and fixed with TCA allowing both a nuclear counting and/or SRB readout. Whole wells were imaged using epi-fluorescence and the number of nuclei was determined using a custom-made ImagePro macro. Plates were further processed for SRB staining as described earlier (Zhang *et al*, 2011). SRB data showed a complete overlap with the nuclear count so this measure was used for normalization. All values are normalized to nuclear count. EOR for control and each gene knockdown is computed by dividing the corresponding ECAR and OCR values. A two-sided *t*-test is applied to examine significant changes between control and knockdown-induced EOR.

**Supplementary information** for this article is available online: http://msb.embopress.org

## Acknowledgements

We would like to thank Hans de Bont and Michiel Fokkelman for their technical support, Yoav Teboulle, Matthew Oberhardt, Edoardo Gaude, Gideon Y. Stein and Tami Geiger for their helpful comments on the manuscript. KY is partially supported by a fellowship from the Edmond J. Safra Bioinformatics center at Tel-Aviv University and is grateful to the Azrieli Foundation for the award of an Azrieli Fellowship; SLD is supported by the Netherlands Consortium for Systems Biology and the EU FP7 Systems Microscopy NoE project (258068) and BvdW from the Netherlands Genomics Initiative. ER acknowledges the generous support of grants from the Israeli Science Foundation (ISF), the Israeli Cancer Research Fund (ICRF) and the I-CORE Program of the Planning and Budgeting Committee and The Israel Science Foundation (grant No 41/11).

## Author contributions

KY and ER conceived and designed the research. SLD, VCB, CF, and BvW designed the experimental procedures. FB and AS contributed the lactate secretion data. KY performed the computational analysis and the statistical

computations. SLD, VMR and VCB performed the experimental procedures. KY, SLD, BvW, and ER wrote the paper.

## Conflict of interest

The authors declare that they have no conflict of interest.

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
