## [Review Process File · Molecular Systems Biology]

A computational study of the Warburg effect identifies metabolic targets inhibiting cancer migration

Keren Yizhak, Sylvia E. Le Dévédec, Vasiliki Maria Rogkoti, Franziska Baenke, Vincent C. de Boer, Christian Frezza, Almut Schulze, Bob van de Water, Eytan Ruppin

Corresponding author: Keren Yizhak, Tel-Aviv University

Review timeline:

Submission date:	18 November 2013
Editorial Decision:	19 December 2013
Revision received:	03 June 2014
Editorial Decision:	03 July 2014
Accepted:	07 July 2014

Editor: Maria Polychronidou

Transaction Report:

1st Editorial Decision

19 December 2013

Thank you again for submitting your work to Molecular Systems Biology. We have now heard back from the three referees who agreed to evaluate your manuscript. As you will see from the reports below, the referees acknowledge that you address a potentially interesting topic. However, they raise a series of concerns, which should be carefully addressed in a revision of the manuscript.

Without repeating all the points listed below, some of the more fundamental issues are the following:

- The advantages of using PRIME and the resulting genome-scale metabolic models need to be convincingly demonstrated.
- Additional experimentation is required to demonstrate that silencing of the predicted targets differentially affects migration and proliferation and that this effect is linked to the AFR. Moreover, extending these results by including additional cell lines will strengthen the impact of the presented conclusions.
- As reviewers #1 and #2 point out, the data regarding the predictive capacity of AFR for anti-cancer drugs need to be presented and discussed in detail.

On a more editorial level, we would like to encourage you to include the source data for the figures that contain essential quantitative information.

 REFEREE REPORTS

Reviewer #1:

The manuscript uses omics data to reconstruct metabolic networks of breast cancer. From estimated fluxes, the ratio between glycolytic and OXPHOS ATP is calculated (the AFR). The authors show that highly glycolytic cells (eg with high AFR) tend to be better "migrant". Perturbation of the AFR upon gene knockdown reduces the migration speed. This work is very interesting. The necessity and benefit of the flux estimation based on omics data is still questionable, but the idea of migration-coupled aerobic glycolysis is both original and relevant.

Major comments:

(1) The predicted lactate production rate and AFR depend massively on the glucose and glutamine uptake rates. Were these measured? Were the measurements included in the GSMM prediction? If not, how do the predicted uptake rates correlate with measured rates?

(2) As a measure of quality of the GSMMs obtained with omics data and PRIDE, the authors emphasize how well models predict lactate formation. However, lactate production results from constraining the maximum oxygen uptake rate GSMM based DIRECTLY on lactate secretion. By maximizing growth, the cells are forced to produce lactate to balance NADH or get energy. Isn't this trivial? The result is NOT an achievement of PRIDE or omics-based network reconstruction and can't be used as a benchmark.

(3) I'm puzzled about the capacity of AFR to predict the NCI60 drug response data. This would be of relevance only for drugs that directly affect aerobic glycolysis. Unfortunately, it's not clear which data were used in the current study because the Ross et al Nat Genetics 2000 paper cited in the text didn't use drugs. I guess the authors meant the closely related Scherf paper, which tested >70'000 compounds on the NCI60 panel. If my assumption is correct, I'm really wondering why the authors make a case out of it since most compounds in that Study inhibit growth and not glycolysis. This is in conflict with the main story that AFR correlated with migration and not with growth. I understand

(4) AFR and EOR seem to differ only minimally. ATP from OXPHOS and OUR are equivalent. ECAR and glycolytic ATP differ seemingly only by the fraction of glycolysis that doesn't end in lactic acid. Overall, the difference is apparently marginal. Not surprisingly, the correlations of AFR and EOR with proliferation and migration (figure 2D) are identical. Was it really necessary to come up with a new term?

(5) A main conclusion is that "the overall picture... is that while glycolytic carbon diverted to biosynthetic pathways may support cell proliferation, non-diverted glycolytic carbon supports cell migration and metastasis". Isn't this shockingly trivial? Any biologist would have known/guessed this in advance without much modeling.

(6) It was shown that silencing of the predicted targets attenuates migration. However, the link to AFR is not proven. Please demonstrate that the actual AFR (with measured glucose uptake rates) or EOR were affected by the knock-down. Please provide full information to demonstrate that migration and proliferation are differentially affected upon modulation. The reported references and percentages are not sufficient.

Minor comments:

(1) units of rates must be consistent throughout the text, ideally fmol/cell/h or nmol/10⁶ cells/h. For instance, in figure 1 it should be possible to directly compare predictions with measurements.

(2) growth rates of cell were taken from the literature. Where from? What is the confidence on these data? Were these measured on perfectly identical conditions (compared to the migration assays)?

(3) In figure 1A, a correlation of 0.36 is objectively low, at least in biological terms. Even worse, the slope is far from diagonal: a doubling in the measured lactate production corresponds to a 10%

increase in the simulation. This is a poor performance for a FBA model. How were the p-values calculated? by permutations or assuming some distributions?

(4) In figure 2A, don't use any normalization and add confidence bars resulting from the sampling procedure. It's important to appreciate the quality of the predictions.

Reviewer #2:

The authors carry out constraint based modeling of human metabolism to investigate relationships of metabolism and cancer cell migration. Their model is based on stoichiometry of the network, constraints of fluxes based on gene expression (which captures cell line specificities) and biomass optimization. Modeling cancer metabolism is an important area as is identifying new drug targets. I have some concerns about this paper.

- The major limitation of the paper is the lack of mechanistic insights. The authors develop a model and then knockdown a few metabolic genes, showing that these genes happen to have a correlation with those involved in migration. But without any reason for why, this appears arbitrary. The major finding is the correlation with the siRNA screen but the finding is very preliminary and there are deeper issues discussed below.

- Modeling the Warburg Effect here relies on a flux capacity constraint. However This constraint basically limits the extent of oxygen consumption that can occur by limiting activity in the mitochondria. Unless there is a direct measure of the mitochondrial capacity (or better yet, a measure of the activity in the modeled cells), it is not a justified assumption and the Warburg Effect in these models appears trivial since it is basically an input in the model and not something that emerges from independent assumptions.

Minor points:

- The authors suggestion that drugs that inhibits metastases alone will lead to less toxic side effects has some grounds. However, the authors should also discuss possible adverse effects of inhibition of migration. Especially since migration is used in a broad range of normal physiological process.

- The AFR measure is predictive for 23% of anti-cancer drugs from a public repository. Could the author list the predicted drugs together with their AFR measure and the other two Warburg measures and IC50 values in a supplemental file? Such a list could be helpful for the cancer research community and increase visibility of the manuscript. The statement that high AFR measures corresponds with higher IC50 values to suppress their growth (e.g. proliferation) seems somewhat counterintuitive with fig 2D. Could the authors reformulate this sentence and/or elaborate.

- Feizi et al(1) report a correlation of 0.241 with a p value of 0.008 for the relation between lactate secretion and growth rate on the Jain et al dataset. In current manuscript a correlation of -0.22 with p value 0.09 is reported. Why this difference in p value?

- Supple. Fig S4 contains correlations of predicted fluxes with growth and migration. Why are some very small correlation significant (e.g. R_GR for lactate) significant while other correlations that are rather large are non-significant (e.g. R_GR for KG). Can the authors add how the p-values were calculated for correlations in the methods section.

- In my opinion the siRNA experiments do show that proliferation and migration are dichotomous. Instead there seems to be an overlapping effect. The majority of the siRNA screens also seem to result in reduction of proliferation by about 50% (suppl. Figure 7). Please report p values on this effect. Also since siRNA usually has a limited effect on gene silencing, no lethal phenotype is to be expected. What was the level of suppression of the silenced genes?

- Moving Suppl. Figure 7 next to Figure 5 would make comparison between proliferation and migration more straightforward.

- In their discussion the authors refer to De Bock et al who show that silencing of PFK interferes with migration. De Bock et al however also show that a reduction in proliferation occurs (figure 1J in

Debock et al). Could the authors also make note of this.

- Migration and proliferation are not mutually exclusive(2). Could the authors maybe make note of this in the discussion?

- The Kaplan-Meier curve uses Survival as a clinical end-point. Numerous publicly available breast cancer gene expression datasets use Time-To-Distant-Metastasis. Would this not be a more relevant clinical end point? Why only used one such dataset? Many are available and using more would give more proof to the proposed concepts. Especially using a dataset from another cancer type would provide additional proof.

Reviewer #3:

Yizhak et al present some very nice work where they build genome-scale metabolic models for the express purpose of investigating the relationship between the Warburg effect and cancer cell migration. An essential step in producing these models is the ability for them include lactate secretion, something previous metabolic models have not successfully done. This advance is a significant one for the field of metabolic modeling and cancer, but an outstanding point that the authors would do well to include, is that constraint based modeling can often be misleading in that, while lactate secretion isn't required for growth in the initial human metabolic reconstruction, that could be due to a number of reasons. For this reason, it's not surprising that their Warburg-specific model does not perform well in predicting lactate secretion on the cell lines experimentally measured by Jain et al.

They are able to quantify the Warburg effect through the ratio of glycolytic to oxidative ATP flux (AFR) and show that this ratio, in their model, has nice predictive power when applied to data from the NCI-60 cancer cell lines, as well as the Kaplan-Meier plots of breast cancer patients.

An outstanding question is how their modeling of lactate secretion performs for a non-cancer system. This would be useful because it would help to understand to what extent their model is recapitulating events that are the result of cell culture growth versus the contribution of lactate secretion in cancer. It would also be a nice baseline against which to understand the differences between cancer and non-cancer (transformed-MCF10A) cells (or possibly muscle cells that also perform oxidative phosphorylation).

To mechanistically link their model predictions to cell migration, they select 17 target enzymes for siRNA knockdown. They find that 12 of these knockdowns significantly impair migration. The question of how much attenuated proliferation contributes to the observed reduction in migration, while addressed, is not particularly convincing. But, then again, it is the opinion of this reviewer that the need to select for gene targets that do not affect proliferation is not necessary. Pertaining to that assertion made by the authors, while the rationale that inhibition of metastasis could be an effective strategy as a cancer treatment is reasonable, that it be independent or somehow better than a strategy that does not kill the cells is at this point more a hypothesis that needs more evidentiary support. To this end I would recommend the authors site work supporting their hypothesis or modify the wording as to not present that their strategy is one of established merit or understanding.

I recommend that the experimental work concerning the effect of knockdown of select targets on migration be expanded to more than one cell line. Preferably, 3-5 cell lines would be more convincing that the observed effects are an accurate reflection of the relationship between migration and the Warburg effect. I would suggest that additional cell lines to be tested include a range of accuracy regarding the AFR or lactate secretion as predicted by the model as to better understand both the value and limitation of the model and underlying biology. The number of target siRNAs is appropriate, and it is interesting that ENOS2 demonstrated what appears to be significantly increased migration (though this was not discussed in the paper).

Typo (paragraph above Results): These potential targets are then filtered further to exclude those predicted to result in cell lethality.

Minor point: For equation 1 in the Methods section, the multiplication of S by the flux vector v as described is not actually a dot product, as the product is the column vector 0 , not the scalar value 0 . Instead of $S \cdot v = 0$, the equation should be $Sv = 0$ or $S * v = 0$.

1st Revision - authors' response

03 June 2014

Please find enclosed a revised version of our manuscript entitled "A computational study of the Warburg effect identifies metabolic targets inhibiting cancer migration".

We were pleased to see that the reviewers had found interest in our manuscript and would like to thank them for their comments and insights, which have greatly helped us in further improving the paper. Following their suggestions, the experimental validations and their analyses were significantly extended and various parts of the paper were rewritten accordingly and in response to other comments arising. Addressing the main comments raised by the reviewers, we have now:

1. Extended the experimental part of the study by *examining the effect of the predicted knock-downs in 3 more cell-lines*, in addition to the one originally presented; as reviewed below, the results further strengthen and confirm the findings that have been reported previously.
2. *Experimentally investigated the effects of the top predicted knockdowns on lactate secretion and oxygen consumption rates*, affirming the fundamental assertion predicted in this study that the anti-migratory effects of these knockdowns are indeed associated with a reduction of the EOR (and AFR) ratio, as predicted.
3. *Provided additional clarifications* regarding the suitability of PRIME for studying the Warburg effect, and have further clarified our findings regarding the relation between the AFR and drug response.

The specific comments of the reviewers are addressed below as follows:

Reviewer #1:

1. *The predicted lactate production rate and AFR depend massively on the glucose and glutamine uptake rates. Were these measured? Were the measurements included in the GSMM prediction? If not, how do the predicted uptake rates correlate with measured rates?*

We thank the reviewer for this comment. Indeed, lactate secretion rates are dependent on both the glucose and glutamine uptake rates and those were indeed measured in the Jain et al. study of the NCI60 panel (Jain et al, 2012), which has been a major undertaking. Please note, however, that the main goal of our study is to identify and test new candidate drug targets and accordingly, the lion share of our analysis is focused on various drug response and gene perturbation simulations for which this data obviously does not exist. Hence, *it has been therefore of prime importance to us to develop and test the models ability to capture the basic Warburg-related phenotypes without the incorporation of such uptake and secretion measurements* (one may further add that such uptake data does not yet exist for the HapMap data, and we aimed to treat both cell collections on equal footing as possible). Second, and more specifically, following the reviewer's comment we have now examined the effects of integrating glucose and glutamine uptake rates as inputs to the wild-type models built. Indeed, the resulting predicted lactate secretion rates were found to be highly correlated with their corresponding measured values (Spearman $R = 0.67$, $P\text{-value} = 1.5e-8$). Importantly and reassuringly, the AFR values predicted in both simulation setups (with and without including the uptake/secretion data) are highly correlated (Spearman $R = 0.9$, $P\text{-value} < 1e-10$). This additional analysis is now briefly reported in the Discussion (page 8) as follows: "The lion share of our analysis is focused on the simulations of perturbations where specific uptake rates are not available. Nonetheless, utilizing such uptake measurements can significantly increase the correlation to the measured lactate rates (Spearman correlation $R = 0.67$, $P\text{-value} = 1.5e-8$), suggesting that uptake rates measurements under perturbation states can significantly increase the models' prediction power."

2. *As a measure of quality of the GSMMs obtained with omics data and PRIDE,*

the authors emphasize how well models predict lactate formation. However, lactate production results from constraining the maximum oxygen uptake rate GSMM based DIRECTLY on lactate secretion. By maximizing growth, the cells are forced to produce lactate to balance NADH or get energy. Isn't this trivial? The result is NOT an achievement of PRIDE or omics-based network reconstruction and can't be used as a benchmark.

The reviewer of course correctly asserts that under maximal growth rate and limited oxygen levels, cells are forced to secrete lactate and maintain high levels of glycolysis. However, the generic human model does not manifest this phenomenon. Specifically, when maximizing growth rate in the generic human model, **even when oxygen uptake rate equals zero**, the model does not *have* to secrete lactate to maintain its maximal growth (i.e., the minimal lactate secretion can be zero). This phenomenon was already noted by Shlomi *et al.* (Shlomi *et al.*, 2011). In that study the authors overcame this hurdle by adding enzyme kinetic constraints that led to forced secretion of lactate when the cells maximize their growth. However, as such enzyme kinetic data exists for only a small subset of human enzymes (Shlomi *et al.* have assigned random values sampled from a certain distribution to most of the enzymes), this approach is far from ideal. Remarkably, the PRIME-derived models do manifest forced lactate secretion under maximal growth rate and when oxygen is not the limiting factor, *without requiring the addition of any enzyme kinetic data*. We thank the reviewer for this comment and now more explicitly clarify this issue in the introduction as follows (page 3): "In the context of studying the Warburg effect, the original human metabolic model does not predict forced lactate secretion under maximal biomass production rate, even when oxygen consumption rate equals zero. This renders it unsuitable for studying the Warburg effect as is, as already noted by (Shlomi *et al.*, 2011). While the addition of solvent capacity constraints has been shown to overcome this hurdle in principle (Shlomi *et al.*, 2011), it requires enzymatic kinetic data which is still mostly absent on a genome-scale".

3. *I'm puzzled about the capacity of AFR to predict the NCI60 drug response data. This would be of relevance only for drugs that directly affect aerobic glycolysis. Unfortunately, it's not clear which data were used in the current study because the Ross *et al* Nat Genetics 2000 paper cited in the text didn't use drugs. I guess the authors meant the closely related Scherf paper, which tested >70'000 compounds on the NCI60 panel. If my assumption is correct, I'm really wondering why the authors make a case out of it since most compounds in that Study inhibit growth and not glycolysis. This is in conflict with the main story that AFR correlated with migration and not with growth. I understand*

We thank the reviewer for this important remark and would like to further explain this issue. First, we apologize for providing the wrong reference and indeed, the correct reference is (Scherf *et al.*, 2000). This has been fixed now in the main text (page 6). While the entire database indeed includes > 70,000 compounds, in our analysis we used the set of 1400 compounds that have been tested at least four times on all or most of the 60 cell lines, similar to the set analyzed in Scherf *et al.* This dataset is referred to as the Drug activity data and was downloaded from the following NCI website: <http://discover.nci.nih.gov/nature2000/natureintromain.jsp>. This information has now been added to the Methods part of the paper under 'Datasets' (page 11).

Second, turning to address the reviewer's question, it is important to note that the targets of most compounds in this database are actually unknown. In fact, Scherf *et al.* have focused on a subset of these compounds that included only 118 compounds whose mechanism of action are putatively understood, and those compounds indeed effect cell growth by various mechanisms. Yet, in the absence of this target information we still find that the Gi50 values of almost third of the compounds across the NCI-60 cell-lines correlates with the AFR levels computed for these cell-lines (in the wild-type models). Out of these, remarkably, *97% are positively correlated with the AFR*, suggesting that the more *'Warburgian' cell-lines are less responsive and therefore require higher dosage of compound to suppress their growth*. Answering the reviewer's question, these findings are not in conflict since the emerging picture implied is that higher AFR cell-lines have lower proliferation rates (see Figure 2D) and hence are less response to anti-proliferative treatments (have higher Gi50 values). Similar results showing that slowly proliferating cells are more resistant to treatment have already been reported in the literature (see main text quote below). To sum this up, the following image portrays the relation between the three measures:

Finally, note that the response to many compounds in this dataset shows a significant association with the AFR measure while having no association with the cells' growth rate. 133 such compounds were identified (Supplementary Dataset S3), possibly suggesting that their mechanism might be related to the Warburg level of the cells rather than to their proliferation.

Accordingly, the corresponding part in the paper has been updated as follows (page 5): "Testing both measures using a genome-wide NCI-60 drug response dataset (Scherf et al, 2000) we find that the model-predicted wild-type AFR levels across all cell-line models are significantly correlated (Spearman P-value < 0.05; FDR corrected with $\alpha = 0.05$) with Gi50 values of 30% of the compounds across these cell-lines (empiric P-value < $9.9e-4$), whereas the model predicted EOR measure accomplish this task for only 19% of the compounds (Methods). Interestingly, we find that out of the 30% AFR-Gi50-correlated compounds, 97% are positively correlated, suggesting that the more 'Warburgian' cell-lines are less responsive and therefore require higher dosage of compound to suppress their growth. The effect of most of these compounds is also negatively correlated to the cells' growth rates, suggesting that slowly proliferating cells are more resistant to treatment (similar results were previously shown for compounds targeting cell growth (Penault-Llorca et al, 2009; Vincent-Salomon et al, 2004)). Interestingly, the response to many compounds in this dataset shows a significant association with the AFR measure while having no association with the cells' growth rate. 133 such compounds were identified (Supplementary Dataset S3), possibly suggesting that their mechanism might be related to the Warburg level of the cells rather than to their proliferation."

4. *AFR and EOR seem to differ only minimally. ATP from OXPHOS and OUR are equivalent. ECAR and glycolytic ATP differ seemingly only by the fraction of glycolysis that doesn't end in lactic acid. Overall, the difference is apparently marginal. Not surprisingly, the correlations of AFR and EOR with proliferation and migration (figure 2D) are identical. Was it really necessary to come up with a new term?*

We introduced the AFR obviously not because we had a desire to come up with a new term but simply because we believe that it is *conceptually* the correct index for quantifying the Warburg level of cells, as it directly quantifies the tradeoff between the glycolytic and oxidative ATP source. In current experimental setups, lactate secretion and oxygen consumption rates are mainly used simply because they are easy to measure, while measuring ATP production in the different compartments requires a much more complex tracer analysis setup. However, metabolic modeling provides an opportunity to define and estimate the more accurate and direct measure of the Warburg effect with no cost, which we find to be an elegant step worthy of attention. We still find it pertinent to additionally report the more commonly used experimental counterpart measure. Indeed, there is a marked significant correlation between the two measures (Spearman correlation $R = 0.66$, P-value = $2e-8$), but it is still far from perfect, justifying the tracking of both in our minds. Finally, in all predictions made throughout the paper (including drug response, the ability to distinguish between epithelial and mesenchymal cell-lines and migration speed), the AFR actually shows a superior performance, thus making it a better measure for quantifying the Warburg effect.

5. *A main conclusion is that "the overall picture... is that while glycolytic carbon diverted to biosynthetic pathways may support cell proliferation, non-diverted glycolytic carbon supports cell migration and metastasis". Isn't this shockingly trivial? Any biologist would have known/guessed this in advance without much modeling.*

The fact that glycolytic carbon that is diverted to biosynthetic pathways is correlated with cell proliferation is indeed quite straightforward and was already shown in several studies, e.g., (Possemato et al, 2011). However, we are not aware of any study showing directly that glycolytic flux (i.e., carbon that is not diverted to pathways branching from the glycolysis) is correlated with

cell migration, though there have been some indications for such an association (e.g., the association of lactate secretion with a high incidence of distant metastasis (Hirschhaeuser et al, 2011). This important observation is the essence of the assertion mentioned above, as we hope is clearer now.

6. *It was shown that silencing of the predicted targets attenuates migration. However, the link to AFR is not proven. Please demonstrate that the actual AFR (with measured glucose uptake rates) or EOR were affected by the knock-down. Please provide full information to demonstrate that migration and proliferation are differentially affected upon modulation. The reported references and percentages are not sufficient.*

We thank the reviewer for this suggestion. Following this comment we conducted a new rounds of experiments aimed precisely at examining the EOR (ECAR/OCR) changes following a selected subset of our top predicted gene knockdowns, those showing the most efficient reduction in cellular migration. In all cases we find a significant reduction in EOR levels, which is also in agreement with the predicted reduction in AFR levels. An entirely new section describing these results was added to the paper ('ECAR and OCR levels following selected gene silencing', pages 7-8) as follows: "To further study the association between reduced AFR levels and impaired cell migration we used the Seahorse XF96 extracellular flux analyser to measure both ECAR and OCR fluxes in the MDA-MB-231 cell-line, following knockdown of a selected group of targets (Methods and Supplementary Figure S6). As the AFR measure is very difficult to measure experimentally, we tested the conventionally measured EOR (ECAR/OCR) as its proxy. We focused on a subset of 7 genes (Figure 5) whose knockdown is predicted to have the highest effect on cell migration and span all three predicted metabolic pathways. As shown in figure 5, a significant EOR reduction versus the control is found for all seven examined genes (two-sided T-test P-value < 0.05, FDR corrected with $\alpha = 0.05$, Methods and Supplementary Table S4). The silencing of the four glycolytic genes (HK2, PGAM1, PGK2, and GAPDH) results in both decreased ECAR and increased OCR levels, while the silencing of the serine- and methionine-associated genes (PSPH, AHCY and PHGDH) results with decreased ECAR solely (Figure 5A). Furthermore, a matching significant difference in experimentally measured EOR levels is found between the lowest and highest AFR reducing genes (one-sided Wilcoxon p-value = 0.05). Overall, taken together our results testify that, as predicted, the knockdown of the top ranked genes results in attenuated cell migration that is accompanied by reduced EOR and AFR levels."

The full information regarding migration and proliferation changes induced by the predicted gene knockdowns (including their significance levels) now appears in Figure 4 and Supplementary Dataset S4.

7. *units of rates must be consistent throughout the text, ideally fmol/cell/h or nmol/10⁶ cells/h. For instance, in figure 1 it should be possible to directly compare predictions with measurements.*

Figure # 1 has been updated accordingly with measured values having the same units of fmol/cell/h. As explained above, measured uptake rates are not considered and therefore the models' units are given as arbitrary units.

8. *growth rates of cell were taken from the literature. Where from? What is the confidence on these data? Were these measured on perfectly identical conditions (compared to the migration assays)?*

The growth rate measurements were taken from an official website of the NCI: http://dtp.nci.nih.gov/docs/misc/common_files/cell_list.html. This information is now added to the Methods part of the paper under 'Datasets' (page 12). These growth rates were confirmed in several studies, including (Jain et al, 2012). The cells in both experiments were grown under the RPMI medium.

9. *In figure 1A, a correlation of 0.36 is objectively low, at least in biological terms. Even worse, the slope is far from diagonal: a doubling in the measured lactate production corresponds to a 10% increase in the simulation. This is a poor performance for a FBA model. How were the p-values calculated? by permutations or assuming some distributions?*

The reviewer is correct that the correlation is not strong; however, as for the reasons explained earlier, glucose and/or glutamine uptake rates were not taken into consideration in this simulation (of note, if these uptake rates are considered, the correlation rises to ~ 0.7). As for the slope, indeed, we are the first to acknowledge that GSMM models are many times not sufficiently strong to make exact quantitative predictions and their key value is in more qualitative and relative predictions (and see the Discussion, page 8). Finally, we computed the Spearman rank correlation using the standard and widely used function 'corr' in Matlab. In this analysis, the correlation p-value is computed using the exact permutation distribution. This information was added to the Methods part of the paper (page 10).

10. *In figure 2A, don't use any normalization and add confidence bars resulting from the sampling procedure. It's important to appreciate the quality of the predictions.*

We thank the reviewer for this comment and following his/her request we added error bars to figure 2A. We however chose to keep the axes of this figure normalized for the following reason: the aim of this figure and study in general, is to present the differences between the cells, rather than the absolute flux levels, acknowledging the known limitations of GSMM models (and see response to the previous comment). The PRIME method changes the maximal flux capacity bounds on the model's reactions such that the bound of each reaction is normalized across the collection of cells. Therefore, the flux rates are meaningful only when compared across the cells and not necessarily in their absolute values. We hence think that presenting the absolute flux values would be presumptuous and misleading and choose to use the normalized values in this plot. Importantly, the normalized AFR and the AFR have a perfect correlation so qualitatively the plot would look identical in both cases. Following the reviewer's comment we specifically expand and relate to this issue when discussing the model's limitations (Discussion, page 8) as follows: "Of note, our modelling approach relies on gene expression differences between the cells and does not take into account specific uptake rates. It is therefore more suited for capturing qualitative rather than exact quantitative differences between the cells, as demonstrated throughout the paper."

Reviewer #2:

1. *The major limitation of the paper is the lack of mechanistic insights. The authors develop a model and then knockdown a few metabolic genes, showing that these genes happen to have a correlation with those involved in migration. But without any reason for why, this appears arbitrary. The major finding is the correlation with the siRNA screen but the finding is very preliminary and there are deeper issues discussed below.*

We thank the reviewer for this important comment. Following the association found between AFR levels and cell migration, we searched for gene knockdowns that will reduce this ratio. Examining all 17 predicted gene targets in 4 breast and lung cancer cell-lines we find that up to 14 of these targets significantly attenuate cell migration (see updated figure 4). Following the reviewer's comment we worked to further establish a mechanistic insight and link these two phenomena. To this end we have conducted a new set of experiments using the Seahorse setup to measure the changes in ECAR and OCR levels that are induced by a selected subset of our top predicted gene knockdowns (those showing the largest effect on cellular migration), and computed the ECAR/EOR ratio (EOR) as an experimental surrogate for the predicted AFR measure (which is yet difficult to measure directly). In all cases we find a significant reduction in EOR levels, in agreement with the predicted reduction in AFR levels. That is, addressing the reviewer's comment, we now have established a direct mechanistic link between the reduction of 'Warburgian' pattern of flux as predicted via AFR levels and measured via ECAR/EOR levels, and the anti-migratory effects of the top predicted perturbations studied here. A new section describing these results was added to the paper ('ECAR and OCR levels following selected gene silencing', pages 7-8) as follows: "To further study the association between reduced AFR levels and impaired cell migration we used the Seahorse XF96 extracellular flux analyser to measure both ECAR and OCR fluxes in the MDA-MB-231 cell-line, following knockdown of a selected group of targets (Methods and Supplementary Figure S6). As the AFR measure is very difficult to measure experimentally, we tested the conventionally measured EOR (ECAR/OCR) as its proxy. We focused on a subset of 7 genes (Figure 5) whose knockdown is predicted to have the highest effect on cell migration and span all

three predicted metabolic pathways. As shown in figure 5, a significant EOR reduction versus the control is found for all seven examined genes (two-sided T-test P-value < 0.05, FDR corrected with $q = 0.05$, Methods and Supplementary Table S4). The silencing of the four glycolytic genes (HK2, PGAM1, PGK2, and GAPDH) results in both decreased ECAR and increased OCR levels, while the silencing of the serine- and methionine-associated genes (PSPH, AHCY and PHGDH) results with decreased ECAR solely (Figure 5A). Furthermore, a matching significant difference in experimentally measured EOR levels is found between the lowest and highest AFR reducing genes (one-sided Wilcoxon p-value = 0.05). Overall, taken together our results testify that, as predicted, the knockdown of the top ranked genes results in attenuated cell migration that is accompanied by reduced EOR and AFR levels.”

2. *Modeling the Warburg Effect here relies on a flux capacity constraint. However This constraint basically limits the extent of oxygen consumption that can occur by limiting activity in the mitochondria. Unless there is a direct measure of the mitochondrial capacity (or better yet, a measure of the activity in the modeled cells), it is not a justified assumption and the Warburg Effect in these models appears trivial since it is basically an input in the model and not something that emerges from independent assumptions.*

We thank the reviewer for this important comment. While capturing the Warburg effect by limiting the oxygen uptake may seem a trivial task, we and others have found that the generic human metabolic model does not capture this phenomenon. Namely, when maximizing growth rate in the generic human model, **even when oxygen uptake rate equals zero**, the model doesn't have to secrete lactate to maintain its maximal growth (i.e., the minimal lactate secretion is zero). This phenomenon was already noted by Shlomi et al. (Shlomi et al, 2011), and these authors overcame this hurdle by adding kinetic constraints to the original stoichiometric information, showing that with this addition lactate secretion is indeed enforced under maximal growth rate. However, as such enzyme kinetic data exists for only a small subset of human enzymes (Shlomi et al. have randomly assigned values to most of the enzymes from a given distribution), this approach is far from ideal. Given this background of previous work we hence find it quite remarkable that the PRIME-derived cancer models do successfully capture the Warburg effect and show enforced lactate secretion under maximal growth rate without the utilization of any kinetic parameters. Importantly, PRIME-derived models that were built here for normal lymphoblasts cell-lines do not show the Warburg effect, (that is, do not have forced lactate secretion even when oxygen uptake rate equals zero). The observed phenomenon is therefore a result of the overall flux capacity constraints set by PRIME in accordance with the unique gene expression signatures of each cell-line and not a consequence of oxygen availability alone.

We now relate to these issues in the Introduction as follows (page 3): "In the context of studying the Warburg effect, the original human metabolic model predicts zero lactate secretion under maximal biomass production rate, even when oxygen consumption rate equals zero. It therefore misses this seemingly trivial biological phenomenon, rendering it unsuitable for studying this phenomenon. While the addition of solvent capacity constraints has been shown to overcome this hurdle in principle, it requires enzymatic kinetic data which is still mostly absent on a genome-scale".

3. *The authors suggestion that drugs that inhibits metastases alone will lead to less toxic side effects has some grounds. However, the authors should also discuss possible adverse effects of inhibition of migration. Especially since migration is used in a broad range of normal physiological process.*

We thank the reviewer for this remark. Accordingly, we have now added the following section to the Discussion (page 9): "It is important to note however that while the inhibition of migration alone might be a good strategy for avoiding the adverse side effects of cytotoxic treatment, cell migration is a crucial process also in normal physiology, for instance, in immune response and tissue repair (Förster et al, 1999; Ridley et al, 2003). Therefore, future anti-migratory drugs may pose different drug selectivity challenges that should be carefully addressed in future studies."

4. *The AFR measure is predictive for 23% of anti-cancer drugs from a public repository. Could the author list the predicted drugs together with their AFR measure and the other two Warburg measures and IC50 values in a supplemental file? Such a list could be helpful for the cancer research community and increase visibility of the manuscript.*

Thanks. The list requested has now been added to the manuscript as Supplementary Dataset S3, and pointed out from the main text (page 5).

5. *The statement that high AFR measures corresponds with higher IC50 values to suppress their growth (e.g. proliferation) seems somewhat counterintuitive with fig 2D. Could the authors reformulate this sentence and/or elaborate.*

This issue might indeed be confusing at first and we thank the reviewer for drawing our attention to this. As displayed in Figure 2D, the AFR is significantly and negatively correlated with cancer proliferation. Our findings show that the Gi50 values of almost third of the compounds across the NCI-60 cell-lines are correlated with the AFR levels computed for these cell-lines (in the wild-type models). Out of these, 97% are *positively* correlated with the AFR, suggesting that the more 'Warburgian' cell-lines are less responsive and therefore require higher dosage of compound to suppress their growth. These findings are not in conflict since the emerging picture implied is that higher AFR cell-lines have lower proliferation rates, and hence are less response to anti-proliferative treatments (have higher Gi50 values). Similar results showing that slowly proliferating cells are more resistant to treatment have already been reported in the literature (see main text quote below). To sum this up, the following image portrays the relation between the three measures:

To better clarify this issue, we have now rewritten the corresponding text as follows (page 5): "Testing both measures using a genome-wide NCI-60 drug response dataset (Scherf et al, 2000) we find that the model-predicted wild-type AFR levels across all cell-line models are significantly correlated (Spearman P-value < 0.05; FDR corrected with $\alpha = 0.05$) with Gi50 values of 30% of the compounds across these cell-lines (empiric P-value < 9.9e-4), whereas the model predicted EOR measure accomplish this task for only 19% of the compounds (Methods). Interestingly, we find that out of the 30% AFR-Gi50-correlated compounds, 97% are positively correlated, suggesting that the more 'Warburgian' cell-lines are less responsive and therefore require higher dosage of compound to suppress their growth. The effect of most of these compounds is also negatively correlated to the cells' growth rates, suggesting that slowly proliferating cells are more resistant to treatment (similar results were previously shown for compounds targeting cell growth (Penault-Llorca et al, 2009; Vincent-Salomon et al, 2004)). Interestingly, the response to many compounds in this dataset shows a significant association with the AFR measure while having no association with the cells' growth rate. 133 such compounds were identified (Supplementary Dataset S3), possibly suggesting that their mechanism might be related to the Warburg level of the cells rather than to their proliferation."

6. *Feizi et al(1) report a correlation of 0.241 with a p value of 0.008 for the relation between lactate secretion and growth rate on the Jain et al dataset. In current manuscript a correlation of -0.22 with p value 0.09 is reported. Why this difference in p value?*

Indeed, Feizi et al. reports on a correlation of -0.241 and we find a correlation of -0.22 (rather marginal differences). We do not know why this difference in the correlation p-value. We computed the Spearman rank correlation using the function 'corr' in Matlab and applied it to the growth rate measurements downloaded from an official website of the NCI:

http://dtp.nci.nih.gov/docs/misc/common_files/cell_list.html (this information is now added to the Methods part of the paper under 'Datasets' (page 12)), and the CORE lactate data that is provided in the Supplementary Material of Jain et al. In this data two values per cell-line are available and therefore the average value for each cell-line was used. The correlation p-value is computed using the exact permutation distribution. This information is now added to the Methods part of the paper

(page 10): "All the correlations reported in the paper are Spearman rank correlation and their associated p-value is computed using the exact permutation distribution."

It might be that Feizi et al. have analyzed the data a bit differently than we report here and it resulted with this minor difference. However, we completely stand behind our computation, which is very basic and was double-checked a few times following the reviewer's comment.

7. *Supple. Fig S4 contains correlations of predicted fluxes with growth and migration. Why are some very small correlation significant (e.g. R_{GR} for lactate) significant while other correlations that are rather large are non-significant (e.g. R_{GR} for α KG). Can the authors add how the p-values were calculated for correlations in the methods section.*

This seeming discrepancy is simply due to the sample size: while we had growth rate measurements for all 60 cell-lines, we had migration speed measurements for only 6 cell-lines. This information was added to the caption of the figure, Supplementary Material (page 8) as follows: "It should be noted that growth rate comparison was done across all 60 cell-lines and the migration comparison for only the 6 available cell-lines." The information how the p-values were calculated for correlations is now provided in the Methods section (page 10).

8. *In my opinion the siRNA experiments do show that proliferation and migration are dichotomous. Instead there seems to be an overlapping effect. The majority of the siRNA screens also seem to result in reduction of proliferation by about 50% (suppl. Figure 7). Please report p values on this effect. Also since siRNA usually has a limited effect on gene silencing, no lethal phenotype is to be expected. What was the level of suppression of the silenced genes?*

Indeed, in our first set of siRNA screen experiments we used inappropriately "mock" transfected cells as control. This negative control implies the use of all the needed reagent for the transfection but it lacks any siRNAs in the mix. We now changed this and relate to a negative control that includes a non-targeting siRNA at the same concentration of the targeting siRNAs (50 nM working concentration). With this more appropriate negative control, we see now that, in general, the knockdown of our selected genes has almost no effect on cell proliferation (except for few genes). The data on the cell migration of the MDA-MB-231 are in agreement with previous data. Furthermore, since the number of cell-lines used for the different assays was now increased from 1 to 4, we used this time the Smart Pools only (a mix of 4 singles) and not the individual singles. Indeed our first screen in the MDA-MB-231 cells experiment used both the singles and the Smart Pools and confirmed that the Smart Pools behave similarly as more than 2 singles. Consequently, we are assured that there are no off-target effects and that the use of the Smart Pools only was sufficient. Finally, we provide now a Western Blot (Supplementary Figure S5) that demonstrates the efficiency of our knockdown protocol for at least two proteins (Paxillin and GAPDH) in all 4 cell-lines (above 80% knock-down).

The entire information regarding the level of migration/proliferation inhibition and its significance level is provided now in Supplementary Dataset S4 (and is referred from the main text page 7). Both migration and proliferation values are presented now in the main text (Figure 4).

9. *Moving Suppl. Figure 7 next to Figure 5 would make comparison between proliferation and migration more straightforward.*

Thanks. Following, Figure 4 in the main text includes now both the effect on migration and on proliferation across the 4 cell-lines examined in this study.

10. *In their discussion the authors refer to De Bock et al who show that silencing of PFK interferes with migration. De Bock et al however also show that a reduction in proliferation occurs (figure 1J in Debock et al). Could the authors also make note of this.*

Thanks you. A note of this issue has now been added to the Discussion (pages 8-9) as follows: "In addition, silencing of PFKFB3 was shown to suppress cell proliferation in about 50% (De Bock et al, 2013)."

11. *Migration and proliferation are not mutually exclusive(2). Could the authors maybe make note of this in the discussion?*

A note along these lines has now been added to the Discussion (page 9): "Overall, the results presented in this study, as well as findings reported by others (Simpson et al, 2008), suggest that proliferation and migration are not mutually exclusive, and the effect of potential targets on both processes should be carefully examined."

12. *The Kaplan-Meier curve uses Survival as a clinical end-point. Numerous publicly available breast cancer gene expression datasets use Time-To-Distant-Metastasis. Would this not be a more relevant clinical end point? Why only used one such dataset? Many are available and using more would give more proof to the proposed concepts. Especially using a dataset from another cancer type would provide additional proof.*

During the revision process we have found a problem with this specific section and hence decided to remove it from the paper. Specifically, in this section we have built the models of the individual clinical samples based on a set of growth-rate-associated genes that were derived from cell-lines data (the NCI-60 collection). We resorted to that 'proxy' since we obviously did not have proliferation rates information for the clinical samples. However, after a careful re-examination of this proxy procedure we have done now, we have second doubts wherever this assumption really has a solid basis – that is, that the same set of genes would be associated with growth in a clinical setting as well. As this section does not influence any of the core results of the paper (which are focused on identifying and studying new anti-migratory Warburg-linked drug targets), we have now chosen to remove it from the paper all together.

Reviewer #3:

1. *Yizhak et al present some very nice work where they build genome-scale metabolic models for the express purpose of investigating the relationship between the Warburg effect and cancer cell migration. An essential step in producing these models is the ability for them include lactate secretion, something previous metabolic models have not successfully done. This advance is a significant one for the field of metabolic modeling and cancer, but an outstanding point that the authors would do well to include, is that constraint based modeling can often be misleading in that, while lactate secretion isn't required for growth in the initial human metabolic reconstruction, that could be due to a number of reasons. For this reason, it's not surprising that their Warburg-specific model does not perform well in predicting lactate secretion on the cell lines experimentally measured by Jain et al.*

Thanks. We cannot agree more with these observations.

One may also add that analyzing the *raw data* of Jain et al. data shows that there is no significant correlation between the glucose uptake rates and the cell-lines measured growth rate (Spearman $R = -0.09$, P-value = 0.48) and a mildly significant negative correlation between glutamine uptake rates and the cells' growth rate (Spearman $R = -0.25$, P-value = 0.052).

2. *An outstanding question is how their modeling of lactate secretion performs for a non-cancer system. This would be useful because it would help to understand to what extent their model is recapitulating events that are the result of cell culture growth versus the contribution of lactate secretion in cancer. It would also be a nice baseline against which to understand the differences between cancer and non-cancer (transformed-MCF10A) cells (or possibly muscle cells that also perform oxidative phosphorylation).*

We agree that this is indeed a very interesting point. Unfortunately, testing this phenomenon in either MCF10A cells or muscle cells is currently not possible due to limitations of the PRIME method. The PRIME pipeline requires both growth rate measurements and a reasonable number of samples in order to build the models. Even if growth rate measurements for MCF10A cells are available, those cells need to be part of a collection of additional cell-lines, as the method normalizes each reaction's bound across the different samples and cannot operate on a single cell. Simply adding the MCF10A cell-line to the NCI-60 dataset would not work as it would not be

comparable to the other cell-lines due to technical (e.g., platform, experiment conditions etc.) rather than biological differences.

However, relating to the reviewer's comment, we have examined whether normal proliferating lymphoblastoid cell-lines must secrete lactate under maximal biomass production (Choy et al, 2008). To this end we used PRIME to build more than 200 cell-specific models of these cell-lines and assessed their lactate secretion under maximal growth. Surprisingly, we found that none of these cell-lines show forced lactate secretion observed in cancer cells. While the Warburg effect is sometimes refers to highly proliferating cells in general, our results show that this phenomenon is apparently more prominent in cancer cells, at least with regard to the lymphoblastoid cell population studied here. We now added a brief note to this end for the interested reader (page 6), stating that: "Moreover, we found that none of the lymphoblast cell-lines show the forced lactate secretion that is observed in cancer cells. While the Warburg effect is sometimes referred in the literature as occurring in highly proliferating cells in general, our analysis finds that this phenomenon is apparently more prominent in cancer cells, at least with regard to the lymphoblastoid cell population studied here."

3. *To mechanistically link their model predictions to cell migration, they select 17 target enzymes for siRNA knockdown. They find that 12 of these knockdowns significantly impair migration. The question of how much attenuated proliferation contributes to the observed reduction in migration, while addressed, is not particularly convincing. **But, then again, it is the opinion of this reviewer that the need to select for gene targets that do not affect proliferation is not necessary.** Pertaining to that assertion made by the authors, while the rationale that inhibition of metastasis could be an effective strategy as a cancer treatment is reasonable, that it be independent or somehow better than a strategy that does not kill the cells is at this point more a hypothesis that needs more evidentiary support. To this end I would recommend the authors site work supporting their hypothesis or modify the wording as to not present that their strategy is one of established merit or understanding.*

We thank the reviewer for raising this important issue. Following this comment we now explicitly discuss both the issue of the proliferation and migration dichotomy and the potential pitfalls of inhibiting cancer migration in the Discussion part of the paper, in lines of his suggestions (page 9): "Overall, the results presented in this study, as well as findings reported by others (Simpson et al, 2008), suggest that proliferation and migration are not mutually exclusive, and the effect of potential targets on both processes should be carefully examined."; "It is important to note however that while the inhibition of migration alone might be a good strategy for avoiding the adverse side effects of cytotoxic treatment, cell migration is a crucial process also in normal physiology, for instance, in immune response and tissue repair (Förster et al, 1999; Ridley et al, 2003). Therefore, the hazard of potential side effects applies for this type of drugs as well and should be carefully addressed in future studies."

4. *I recommend that the experimental work concerning the effect of knockdown of select targets on migration be expanded to more than one cell line. Preferably, 3-5 cell lines would be more convincing that the observed effects are an accurate reflection of the relationship between migration and the Warburg effect. I would suggest that additional cell lines to be tested include a range of accuracy regarding the AFR or lactate secretion as predicted by the model as to better understand both the value and limitation of the model and underlying biology. The number of target siRNAs is appropriate, and it is interesting that ENOS2 demonstrated what appears to be significantly increased migration (though this was not discussed in the paper).*

Following the reviewer's suggestion, we now have experimentally examined the knockdown effects of our predicted targets in 4 cell-lines: MDA-MB-231, MDA-MB-435s, BT549 and A549, that have different predicted AFR levels. Encouragingly, 8-13 out of the 17 predicted targets (depends on the cell-line tested) were found to significantly attenuate cell migration in these cell-lines. The entire set of the results obtained is provided now in Supplementary Dataset S4, Figure 4 and the corresponding part in the main text (page 7) has been updated as follows: "To experimentally test our predictions we silenced the 17 predicted AFR-reducing genes and examined their phenotypic effects in the MDA-MB-231, MDA-MB-435, BT549 and A549 cell-lines. Knockdown experiments were performed with Smart Pools from Dharmacon using a live cell migration and fixed proliferation assays (Methods). 8-13 out of the 17 enzymes (8-10 out of 12 metabolic reactions)

were found to significantly attenuate migration speed in each cell-line (two-sided T-test P-value < 0.05, FDR corrected with $q = 0.05$, Figure 4, Methods and Supplementary Dataset S4). This result is highly significant as only 17% of the metabolic genes were found to impair cell migration in a siRNA screen of 190 metabolic genes (Fokkelman M., Rogkoti VM et al., unpublished data, Bernoulli P-value in the range of $3.9e-3$ and $1.18e-7$). Of note, the association between the gene expression of the predicted targets and the measured migration speed is insignificant for all targets but one, testifying for the inherent value of our model-based prediction analysis (Supplementary Table S3). It should also be noted that the knockdown of the three splices of the enolase gene have almost no significant effect on these cells' migration speed, possibly because of isoenzymes backup mechanisms. Importantly, most of the gene knockdown experiments do not manifest any significant effects on cell proliferation (Figure 4). In accordance with the findings of Simpson et al. (Simpson et al, 2008), we found that the correlation between the reduction in migration speed and reduction in proliferation rate is mostly insignificant (Supplementary Dataset S4), suggesting that the reduced migration observed is not simply a consequence of common mechanisms hindering proliferation, but rather that it occurs due to the disruption of distinct migratory-associated metabolic pathways."

Furthermore, we also examined the EOR (ECAR/OCR) changes following a selected subset of our predicted gene knockdowns. In all cases we found a significant reduction in EOR levels, which was in agreement with the predicted reduction in AFR levels. A new section describing these results was added to the paper ('ECAR and OCR levels following selected gene silencing', pages 7-8) as follows: "To further study the association between reduced AFR levels and impaired cell migration we used the Seahorse XF96 extracellular flux analyser to measure both ECAR and OCR fluxes in the MDA-MB-231 cell-line, following knockdown of a selected group of targets (Methods and Supplementary Figure S6). As the AFR measure is very difficult to measure experimentally, we tested the conventionally measured EOR (ECAR/OCR) as its proxy. We focused on a subset of 7 genes (Figure 5) whose knockdown is predicted to have the highest effect on cell migration and span all three predicted metabolic pathways. As shown in figure 5, a significant EOR reduction versus the control is found for all seven examined genes (two-sided T-test P-value < 0.05, FDR corrected with $q = 0.05$, Methods and Supplementary Table S4). The silencing of the four glycolytic genes (HK2, PGAM1, PGK2, and GAPDH) results in both decreased ECAR and increased OCR levels, while the silencing of the serine- and methionine-associated genes (PSPH, AHCY and PHGDH) results with decreased ECAR solely (Figure 5A). Furthermore, a matching significant difference in experimentally measured EOR levels is found between the lowest and highest AFR reducing genes (one-sided Wilcoxon p-value = 0.05). Overall, taken together our results testify that, as predicted, the knockdown of the top ranked genes results in attenuated cell migration that is accompanied by reduced EOR and AFR levels."

5. *Typo (paragraph above Results): These potential targets are then filtered further to exclude those predicted to result in cell lethality.*

Thanks. This typo has been fixed (page 3).

6. *Minor point: For equation 1 in the Methods section, the multiplication of S by the flux vector v as described is not actually a dot product, as the product is the column vector 0, not the scalar value 0. Instead of $S \cdot v = 0$, the equation should be $Sv = 0$ or $S * v = 0$.*

Thanks. This has been fixed accordingly (page 10).

References

Choy E, Yelensky R, Bonakdar S, Plenge RM, Saxena R, De Jager PL, Shaw SY, Wolfish CS, Slavik JM, Cotsapas C, Rivas M, Dermitzakis ET, Cahir-McFarland E, Kieff E, Hafler D, Daly MJ, Altshuler D (2008) Genetic Analysis of Human Traits In Vitro: Drug Response and Gene Expression in Lymphoblastoid Cell Lines. *PLoS Genet* **4**: e1000287

De Bock K, Georgiadou M, Schoors S, Kuchnio A, Wong Brian W, Cantelmo Anna R, Quaegebeur A, Ghesquière B, Cauwenberghs S, Eelen G, Phng L-K, Betz I, Tembuyser B, Brepoels K, Welti J, Geudens I, Segura I, Cruys B, Bifari F, Decimo I et al (2013) Role of PFKFB3-Driven Glycolysis in Vessel Sprouting. *Cell* **154**: 651-663

Dolfi S, Chan L, Qiu J, Tedeschi P, Bertino J, K. H, Oltvai Z, Vazquez A (2013) The metabolic demands of cancer cells are coupled to their size and protein synthesis rates. *Cancer & Metabolism* **1**

Förster R, Schubel A, Breitfeld D, Kremmer E, Renner-Müller I, Wolf E, Lipp M (1999) CCR7 Coordinates the Primary Immune Response by Establishing Functional Microenvironments in Secondary Lymphoid Organs. *Cell* **99**: 23-33

Hirschhaeuser F, Sattler UGA, Mueller-Klieser W (2011) Lactate: A Metabolic Key Player in Cancer. *Cancer Research* **71**: 6921-6925

Jain M, Nilsson R, Sharma S, Madhusudhan N, Kitami T, Souza AL, Kafri R, Kirschner MW, Clish CB, Mootha VK (2012) Metabolite Profiling Identifies a Key Role for Glycine in Rapid Cancer Cell Proliferation. *Science* **336**: 1040-1044

Penault-Llorca F, André F, Sagan C, Lacroix-Triki M, Denoux Y, Verrièle V, Jacquemier J, Baranzelli MC, Bibeau F, Antoine M, Lagarde N, Martin A-L, Asselain B, Roché H (2009) Ki67 Expression and Docetaxel Efficacy in Patients With Estrogen Receptor–Positive Breast Cancer. *Journal of Clinical Oncology* **27**: 2809-2815

Possemato R, Marks K, Shaul Y, Pacold M, Kim D, Birsoy K, Sethumadhavan S, Woo H, Jang H, Jha A, Chen W, Barrett F, Stransky N, Tsun Z, Cowley G, Barretina J, Kalaany N, Hsu P, Ottina K, Chan A et al (2011) Functional genomics reveal that the serine synthesis pathway is essential in breast cancer. *Nature* **476**: 346 - 350

Ridley AJ, Schwartz MA, Burridge K, Firtel RA, Ginsberg MH, Borisy G, Parsons JT, Horwitz AR (2003) Cell Migration: Integrating Signals from Front to Back. *Science* **302**: 1704-1709

Scherf U, Ross DT, Waltham M, Smith LH, Lee JK, Tanabe L, Kohn KW, Reinhold WC, Myers TG, Andrews DT, Scudiero DA, Eisen MB, Sausville EA, Pommier Y, Botstein D, Brown PO, Weinstein JN (2000) A gene expression database for the molecular pharmacology of cancer. *Nat Genet* **24**: 236-244

Shlomi T, Benyamini T, Gottlieb E, Sharan R, Ruppin E (2011) Genome-Scale Metabolic Modeling Elucidates the Role of Proliferative Adaptation in Causing the Warburg Effect. *PLoS Comput Biol* **7**: e1002018

Simpson KJ, Selfors LM, Bui J, Reynolds A, Leake D, Khvorova A, Brugge JS (2008) Identification of genes that regulate epithelial cell migration using an siRNA screening approach. *Nat Cell Biol* **10**: 1027-1038

Vincent-Salomon A, Rousseau A, Jouve M, Beuzebec P, Sigal-Zafrani B, Fréneaux P, Rosty C, Nos C, Campana F, Klijanienko J, Al Ghuzlan A, Sastre-Garau X (2004) Proliferation markers predictive of the pathological response and disease outcome of patients with breast carcinomas treated by anthracycline-based preoperative chemotherapy. *European Journal of Cancer* **40**: 1502-1508

2nd Editorial Decision

03 July 2014

Thank you again for submitting your work to Molecular Systems Biology. We have now heard back from the two referees who agreed to evaluate your manuscript. As you will see from the reports below, the referees are now satisfied with the modifications made and support publication of the work.

REFEREE REPORTS

Reviewer #1:

The authors have exhaustively addressed my criticism. The revised manuscript is sufficiently explicit and objective on all key aspects. Overall, it's a sound study with an important, novel finding.

Reviewer #3:

The revision addressed all of my previous concerns. I believe it is ready now for publication.